



# Development of a Reduced Complexity Plant Canopy Physics Surrogate Model for use in Chemical Transport Models: A Case Study with GEOS-Chem v12.3.0

Sam J. Silva[1,2], Colette L. Heald[1], Alex B. Guenther[3]

[1]Department of Civil and Environmental Engineering, Massachusetts Institute of Technology, Cambridge, MA, USA
[2]Now at: Pacific Northwest National Laboratory, Richland, WA, USA
[3]Department of Earth System Science, University of California Irvine, Irvine, CA, USA

Corresponding Authors: Sam J. Silva (sam.silva@pnnl.gov) and Colette L. Heald
(heald@mit.edu)

**Abstract.** Biosphere-atmosphere interactions strongly influence the chemical composition of the atmosphere. Simulating these interactions at a detailed process-based level has traditionally been computationally intensive and resource prohibitive, commonly due to complexities in calculating radiation and light at the

leaf level within plant canopies. Here we describe a surrogate canopy physics model based on the MEGAN3 detailed canopy model parameterized using a statistical learning technique. This surrogate canopy model is designed specifically to rapidly calculate leaf-level temperature and photosynthetically active radiative (PAR) for use in large-scale chemical transport models (CTMs). Our surrogate model can

reproduce the dominant spatiotemporal variability of the more detailed MEGAN3 canopy model to within 10% across the globe. Implementation of this surrogate model into the GEOS-Chem CTM leads to small local changes in ozone dry deposition velocities of less than 5%, and larger local changes in isoprene emissions of up to ~40%, though annual global isoprene emissions remain largely consistent (within

5%). These changes to surface-atmosphere exchange lead to modest changes in





surface ozone concentrations of ±1 ppbv. The use of this surrogate canopy model

drives emissions of isoprene and concentrations of surface ozone closer to

observationally constrained values, without any noticeable impact on computational

demand. Additionally, this surrogate model allows for the further development and

implementation of leaf-level emission factors in the calculation of biogenic

emissions in the GEOS-Chem CTM. Though not the focus of this work, this ultimately

enables a complete implementation of the MEGAN3 emissions framework within

GEOS-Chem, which produces 570 Tg yr$^{-1}$ of isoprene in 2012.

## 1. Introduction

The biosphere plays an important role in modulating the abundance and variability

of trace gases and aerosol in the atmosphere. Direct emissions of gas phase species

are drivers of the majority of the natural reactivity in the atmosphere, and are

important precursor sources to pollutants and climate relevant species like ozone

and particulate matter (Guenther et al., 2012; IPCC, 2013; Safieddine et al., 2017).

On the other hand, vegetation serves as a direct sink for these same species through

a process known as dry deposition (Lelieveld and Dentener, 2000; Silva and Heald,

2018). The physical structure of the vegetation can also influence the production

and loss of atmospheric constituents through changes to atmospheric turbulent

transport and reductions in the actinic flux below the canopy (e.g. Makar et al.,

2017). Additionally, chemical reactions occurring within the plant canopy act as a

source and sink for reactive species in the above-canopy atmosphere (Goldstein et

al., 2004; Makar et al., 1999). Ultimately, the balance between the role vegetation



plays as a chemical source and sink is a controlling factor for the abundance and variability of trace gases and aerosol across many regions of the globe (e.g. Geddes

et al., 2016; Silva et al., 2016; Unger, 2014). It is thus important to properly account for these processes when simulating the composition and chemistry of the atmosphere.

Explicitly simulating biosphere-atmosphere interactions necessitates a detailed representation of physical, chemical, and biological processes that occur at the scale

of an individual plant. This is typically achieved by integrating a set of energy and radiative balance equations vertically throughout a canopy (e.g. Ashworth et al., 2015, 2016; Goudriaan and Laar, 1994). This sort of physical model of the canopy calculates the environmental parameters that drive the biological and chemical processes which ultimately impact the atmospheric fluxes of trace gases and aerosol

(Guenther et al., 2012; Lamb et al., 1996). These canopy models tend to be computationally quite expensive, and are based on measurements taken at very fine resolution (e.g. meter or less), while most atmospheric chemical transport models operate on the 10-200 km scale. Reconciling these differences in scale and addressing the steep computational requirements inherent in both canopy models

and atmospheric chemical transport models are critical challenges in simulating chemically relevant interactions between the biosphere and the atmosphere.

Given the computational costs, atmospheric chemical transport models approximate canopy physics and the resulting effects on biosphere-atmosphere interactions through various parameterizations. (e.g. Guenther et al., 2006; Wesely, 1989; Zhang



et al., 2003). Most of these parameterizations are based on observed relationships,

and intended to reduce the computational load around the calculation of the

temperature of leaves and the amount of light (specifically photosynthetically active

radiation, PAR) reaching leaves throughout the canopy. These model

parameterizations commonly assume that the temperature of leaves is equal to the

air temperature just above the plant canopy (e.g. Guenther et al., 2006; Millet et al.,

2010) or are based on parameterizations that ignore leaf temperature entirely (e.g.

Wesely, 1989). The parameterizations for leaf level PAR vary widely; from assuming

that the PAR reaching leaves in the canopy is equal to the flux of PAR incident on the

top of the canopy, to having some sort of reduced complexity multiplicative factor

that represents the bulk canopy effects (e.g. shading of leaves, Guenther et al., 2006;

Wang et al., 1998). To our knowledge, the overall impact of these parameterized

assumptions on the fidelity of modern chemical transport models has not been

comprehensively characterized. However for biogenic isoprene emissions, these

canopy approximations can lead to regional differences of greater than 20% relative

to a fully detailed canopy model (Guenther et al., 2006).

Direct representation of these processes is a necessary step to improve model

reliability and validity, particularly in a rapidly changing Earth System (Committee

on the Future of Atmospheric Chemistry Research et al., 2016). Currently, many

processes related to canopy energy and radiative balance are not represented in

models of atmospheric chemistry due to computational constraints. In this work, we

present a reduced complexity canopy model to calculate leaf temperature and PAR

for use in large-scale chemical transport models. This reduced complexity model





removes the need for approximating bulk effects of plant canopies on leaf-level PAR
and leaf temperature, and it allows for a more explicit process-based representation

of these effects on biosphere-atmosphere interactions at the leaf level. Our reduced
model reproduces the output of the more detailed vegetation model well, without
the large computational overhead.

**2. MEGAN3 Canopy Model**

We develop and implement a computationally efficient surrogate of the MEGAN3.0

canopy model (https://bai.ess.uci.edu/megan, last accessed 04/09/2019), an
update from previous versions of MEGAN (Guenther et al., 2006, 2012). This canopy
model calculates leaf temperature and leaf-level PAR for a 5-layer plant canopy for
both sunlit and shaded leaves, where each canopy layer represents a fraction of the
total plant canopy. The model is originally based largely on Goudriaan and Laar

(1994) and a brief description follows; for more information see Guenther et al.
(1999, 2006, 2012).

In the MEGAN3 canopy model the fraction of sunlit leaves in the canopy decreases
exponentially as a function of the local solar elevation angle, canopy leaf area index
(LAI), a clustering coefficient that accounts for leaf geometries, and a canopy

transparency coefficient representing the fraction of the canopy that does not
intercept incident radiation. The leaf temperature is calculated from a system of
energy balance equations based on Goudriaan and Laar, (1994) and Leuning et al.
(1995), with a maximum absolute difference between air temperature and leaf
temperature of 10°C. Leaf-level PAR is computed as a function of incoming radiation



incident to the canopy top, the sunlit fraction of leaves, LAI, and a suite of geometric

and radiative look up table characteristics, predominantly based on Goudriaan and

Laar (1994), Leuning et al. (1995), and Spitters (1986). The full MEGAN3 canopy

model takes as input: time (day and hour), geographical location (latitude and

longitude), air temperature, incident radiation on the top of the canopy, wind speed,

humidity, air pressure, LAI, and a set of canopy characteristics (canopy biomass

distribution, clustering coefficients, etc.) in the form of a look up table that varies by

six vegetation types. The six vegetation types are: needleleaf trees, tropical forest

trees, temperate broadleaf trees, shrubs, herbaceous plants, and crops. It is

important to note that differing canopy model choice and parameter selection can

result in substantial changes to the ultimate estimates of biosphere-atmosphere

exchange (Keenan et al., 2011).

The MEGAN3 canopy model has been specifically developed for use in simulating

biogenic emissions, and has been extensively applied in related studies (e.g. Chen W.

H. et al., 2018; Geron et al., 2016; Guenther et al., 2006). Additionally, the MEGAN

framework has been widely adopted across a variety of regional and global models

(e.g. GEOS-Chem, WRF-CHEM, and CESM). Thus the MEGAN3 canopy model is a

good candidate for surrogate model development because it enables a direct

implementation of improved process-based canopy physics into a variety of 3D

models without the need for substantial model architecture development.

**3. Surrogate Model Development**





To begin the surrogate model development, we first use a variable selection approach to evaluate and rank which of the suite of model input variables are most important for the simulation of both leaf level PAR and temperature. To do this, we use a machine learning regression method for model simplification and

parameterization, specifically LASSO (Least Absolute Shrinkage and Selection Operator, Hastie et al., 2001). As applied here, LASSO is a regression method that calculates linear coefficients through a modified least squares cost function, with the addition of a penalized L1 norm (the sum of the absolute value of the coefficients). While LASSO was originally developed as a complete regression method, we follow

the recommendations of Hastie et al. (2001) and use LASSO only for variable importance ranking and dimensionality reduction of the input variable space to the model.

We apply the linear LASSO method for rankings across a full year of 3-hourly simulated canopy physics from the MEGAN3 canopy model at the global scale for the

year 2012. Input meteorology is from MERRA-2 assimilated meteorological fields at 2°x2.5° horizontal resolution (Gelaro et al., 2017), and the vegetation distribution from the Olson 2001 land maps (Olson et al., 2001). LAI is derived from the MODIS TERRA MOD15A2 Product (Myneni et al., 2002, 2007) re-gridded to 2°x2.5° horizontal resolution and a monthly timescale. This input data is identical to that

used in the GEOS-Chem chemical transport model (CTM) described below for direct comparison with prior work and ease of implementation into that CTM. The spatial distribution of vegetation and LAI are summarized in Figures 1 and 2, respectively. In general, forested land classes have the highest LAI values, and are spread





throughout the tropics and the northern latitudes. Crops, grasses, and shrubs are

located predominantly in transitionary landscapes and near regions of larger

population (e.g. India, Central North America, etc.), and tend to have lower LAI

values.

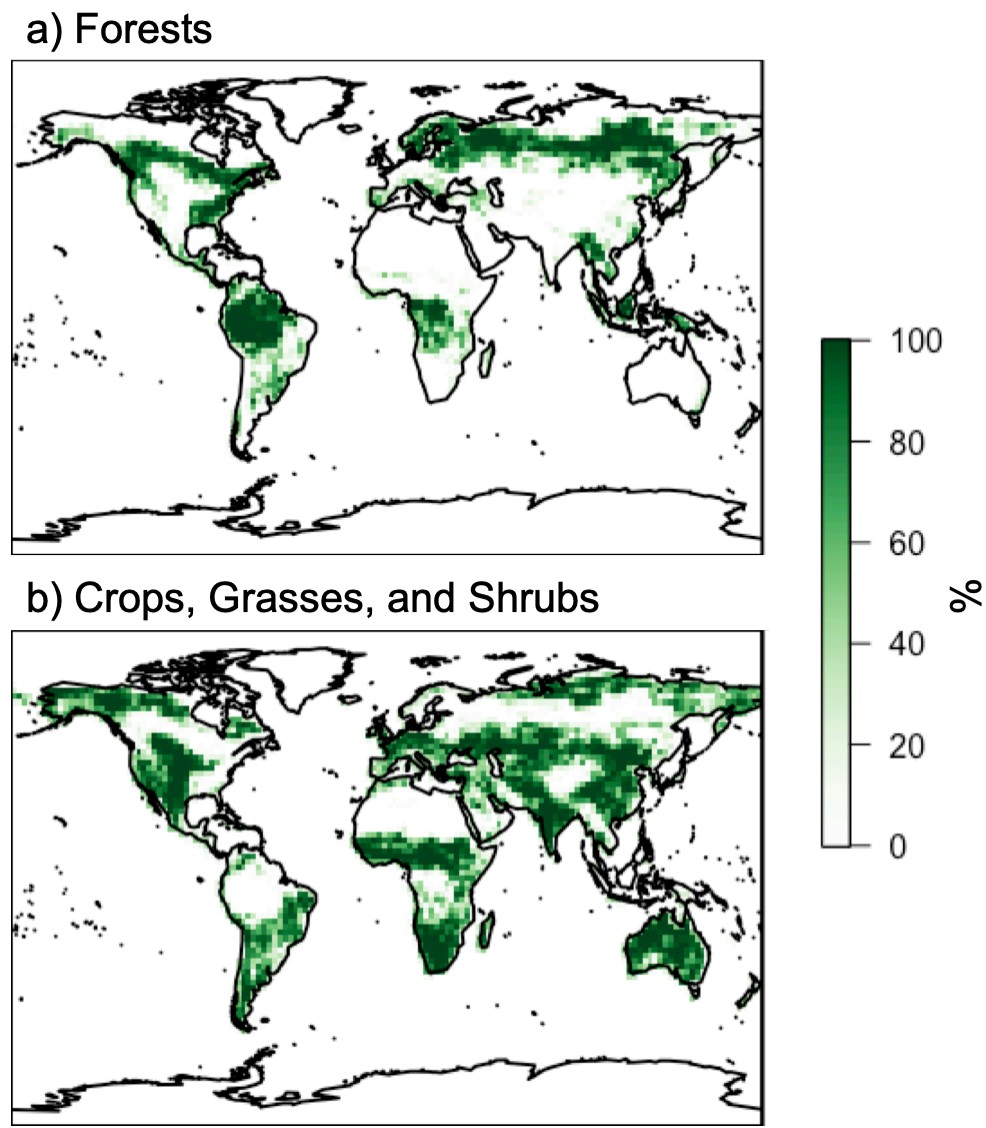





**Figure 1.** The percent of each 2°x2.5° gridbox occupied by each vegetation class

used in this work. Panel a) is forested vegetation, and panel b) is crops, grasses, and

shrubland.

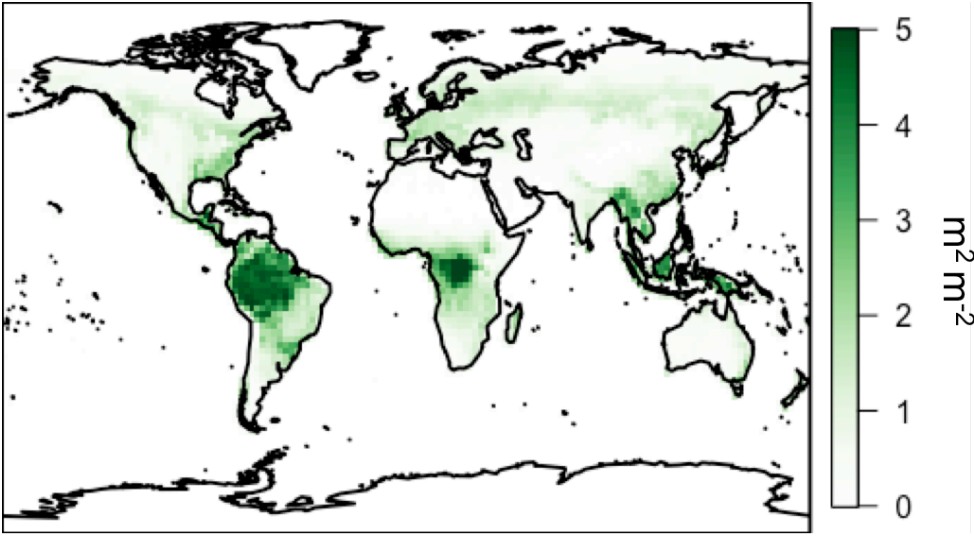

**Figure 2.** Annual Average LAI from MODIS for the year 2012.

The LASSO importance rankings are remarkably consistent for both sunlit and

shaded leaves and for all vertical levels of the canopy. For each quantity, the two

highest ranked variables are consistent at each layer throughout the canopy, and

have substantially larger importance to the final result than any additional variable.

For brevity we discuss only those first two variables here. The two most important

variables for the calculation of leaf temperature are air temperature and wind

speed. Air temperature dominates in importance for the calculation of leaf

temperature, with a larger coefficient emerging at a higher L1 norm penalty

weighting. Other variables that are physically important in nature (e.g. solar





radiation) do not appear important in the LASSO rankings due in part to how they
covary with air temperature, and how the rankings are derived separately for sunlit
and shaded leaves. For the calculation of leaf PAR we find that the two most
important variables are PAR out of the lowermost atmospheric gridbox (incident on
the canopy), and the local vegetation LAI. We use these selected variables to develop
a simplified parameterization for leaf temperature and PAR.

We model leaf temperature for a given canopy level, i ($T_{i,leaf}$, K) as linear with 2-
meter air temperature ($T_{air}$, K):

(1) $T_{i,leaf} = A_i + B_i *T_{air}$

Where $A_i$ and $B_i$ are fitted parameters per canopy level (i). This ignores the addition
of the second most important variable, wind speed. However, the addition of wind
speed to the regression only improves the performance of the model by less than
1% in total bias and $R^2$; thus for simplicity, we neglect this variable.

For the calculation of leaf-level PAR at a given canopy level ($PAR_{i,leaf}$, $\mu$mol m$^{-2}$ s$^{-1}$),
we use an exponential Beers-Law analogue, including the influence of Leaf Area
Index (LAI) and the PAR incident to the top of the canopy ($PAR_{toc}$, W/m$^2$):

(2) $PAR_{i,leaf} = PAR_{toc} * \exp(C_i + D_i*LAI)$.

Where $C_i$ and $D_i$ are fitted parameters per canopy level (i).This exponential
functional form is chosen due to the observed and simulated relationships between



LAI and canopy light interception (Engel et al., 1987; Goudriaan and Monteith,

1990) following a similar functional form.

We fit equations 1 and 2 for all layers of the canopy and for sunlit and shaded leaves,

resulting in 20 total free parameters necessary to model the entire plant canopy

across the globe. In this regression method, we ignore the role of differing

vegetation classes and apply the regression agnostic to vegetation type. This is done

to keep the total necessary number of free parameters low (20 versus 120), and

because this more parsimonious model performs quite well (see Section 3.1)

without the need for additional vegetation type specific coefficients. The resulting

surrogate model coefficients are summarized in Table 1.

| Canopy Level | | A | B | C | D |
|---|---|---|---|---|---|
| Sunlit Leaves | 1 | -13.891 | 1.064 | 1.083 | 0.002 |
| | 2 | -12.322 | 1.057 | 1.096 | -0.128 |
| | 3 | -1.032 | 1.031 | 1.104 | -0.298 |
| | 4 | -5.172 | 1.050 | 1.098 | -0.445 |
| | 5 | -5.589 | 1.051 | 1.090 | -0.535 |
| Shaded Leaves | 1 | -12.846 | 1.060 | 0.871 | 0.015 |
| | 2 | -11.343 | 1.053 | 0.890 | -0.141 |
| | 3 | -1.068 | 1.031 | 0.916 | -0.368 |
| | 4 | -5.551 | 1.051 | 0.941 | -0.592 |
| | 5 | -5.955 | 1.053 | 0.956 | -0.743 |

**Table 1. Regression Coefficients for the Canopy Surrogate Model. Canopy level 1 represents the top of the canopy.**

The final quantity necessary for the canopy model is the fraction of sunlit and

shaded leaves. Here, that fraction in each layer of the plant canopy is calculated

directly following the MEGAN code, (see Guenther et al., 2006, 2012), without any

model simplification.  The sunlit fraction is calculated as follows:



$$(3)\ K_b = 0.5 * \frac{C_1}{\sin\beta} \frac{\alpha_1}{\sin\beta}$$

$$(4)\ Sunlit\ Fraction = \exp\left(K_b * \frac{LAI}{1-\alpha_2} * f\right)$$

Where $\beta$ is the solar angle above the horizon, $K_b$ is the extinction coefficient for

black leaves, $C_1$ is the canopy clustering coefficient, $C_2$ is the canopy transparency,

and $f$ is the fraction of biomass in the canopy light travels through to reach a given

leaf (the vector [0.05, 0.23, 0.5, 0.77, 0.95]). Consistent with the MEGAN3 parent

canopy model, we assume a Gaussian distribution of biomass in the canopy,

centered in the middle canopy layer, and a canopy transparency of 0.2, and a leaf-

clustering coefficient of 0.9.

From this relatively simple three-function parameterization (Leaf Temperature,

Leaf PAR, and Sunlit Leaf fraction), we are able to implement more physically

realistic parameterizations for biosphere-atmosphere interactions.

### 3.1. Surrogate Model Performance

Here, we evaluate the surrogate model performance for all vegetation globally.

### 3.1.1 Temperature

The surrogate model simulated annual canopy average leaf temperature

distribution and performance for 2012 is summarized in Figures 3 and 4. In Figure

3, the average for each canopy layer is calculated as a sum of the sunlit and shaded

leaves, weighted by the sunlit fraction of that layer. In turn, the canopy average is

calculated as the weighted sum of the layer averages, weighted by the fraction of the





canopy biomass in each layer. The annual average temperature is shown in Figure

3a, where it largely follows a latitudinal gradient. The warmest temperatures are

~310 K in the tropics, and the coldest average leaf temperatures are ~280 K in the

northern high latitude boreal regions. The surrogate model is linear with the 2-

meter "near-surface" air temperature, and therefore follows that spatial distribution

directly.

The surrogate model for leaf temperature performs well, with the annual average

spatial $R^2$ and mean bias relative to the full model shown in Figures 3b and 3c,

respectively. Across all regions, the $R^2$ is very high, indicating that a linear

relationship between 2-meter air temperature and canopy average temperature is a

good approximation for capturing the variability of the full MEGAN3 canopy model.

The temperature $R^2$ drops below 0.90 only in coastal regions/gridboxes that contain

very little vegetation, representing less than 5% of all vegetated areas. The

temperature surrogate bias is also generally quite low, as shown in Figure 3c. The

majority of regions have an absolute mean bias of less than 1 K, and more than 90%

of the annual average mean biases are less than 2 K. The surrogate model generally

computes temperatures that are biased cool over highly vegetated tropical and sub-

tropical regions, and slightly overestimates temperature over northern boreal

forests (by ~0.1 K). The most substantial overestimations occur in or near the hot

and arid regions of the globe, and always in regions where there is little vegetative

cover at all. On a relative scale these biases are quite small; all are less than 1% of

the total magnitude.





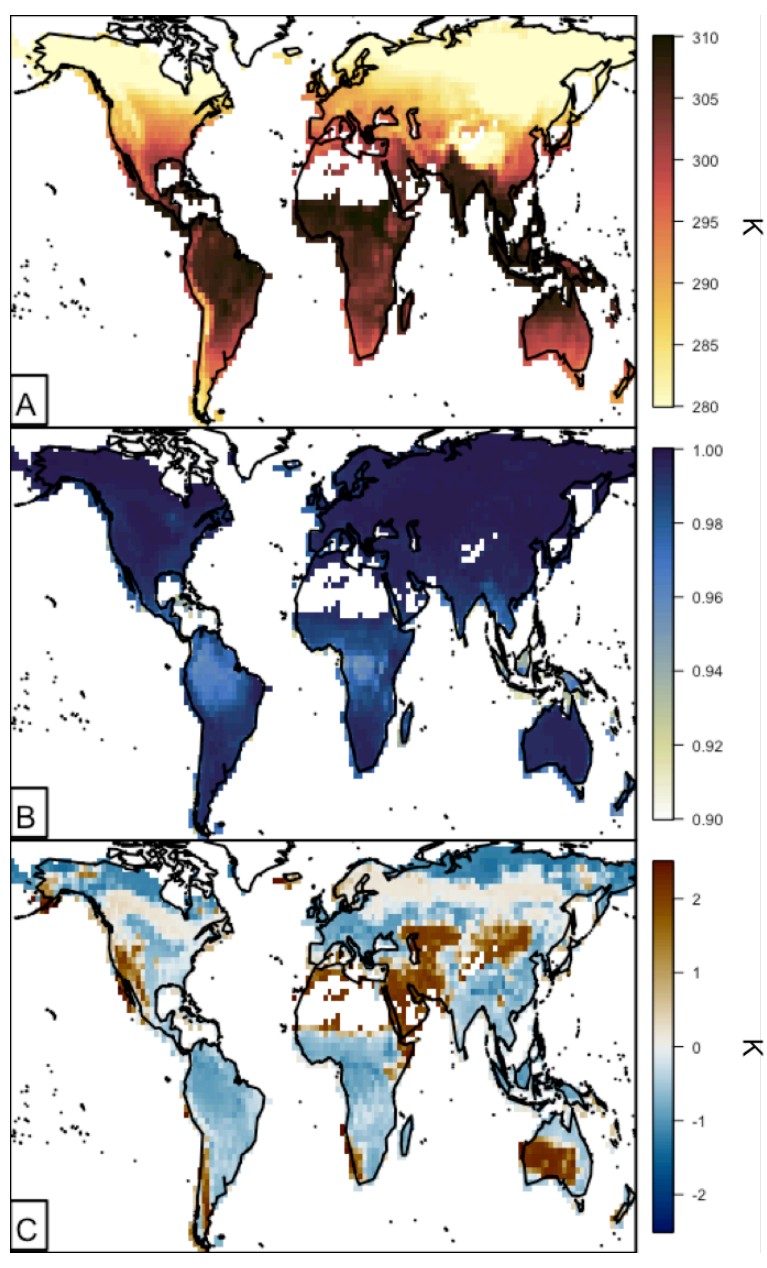

**Figure 3. Annual canopy average spatial temperature surrogate model performance for 2012. Panels are as follows: A) Annual average surrogate model leaf level temperature (Kelvin), B) R² between the surrogate and the full model leaf level temperature, C) Annual average leaf level temperature bias (surrogate-full model, K)**






The vertical profile of annual average leaf temperature is shown in Figure 4a. The broad shape of the vertical distribution is consistent everywhere. The upper canopy layers are cooler than the lower canopy layers, as an insulating effect from air temperature occurs within the canopy. The higher order variability (e.g. small

differences within layers at the top and bottom of the canopy) stems from the more detailed representation of canopy energy balance in the full MEGAN3 model, which includes the influence of terms like PAR, relative humidity, LAI, and wind speed. However, the generally consistent behavior of this higher order variability allows for it to be reproduced in the simplified surrogate model.

Similar to the spatial performance, the overall surrogate model performs well throughout the canopy. The surrogate model temperature $R^2$ is shown in Figure 4b. The values are all near 1.0, with the lowest value of 0.97 in the middle of the canopy, where the transition from cooler to warmer leaves is slightly more difficult for the surrogate model to capture. As demonstrated in Figure 4c, the bias throughout the

canopy is low as well. On a global annual average, the surrogate model is biased cool, but only slightly (on both a relative and absolute scale). The highest magnitude bias is at the top canopy layer, at -0.04 K.





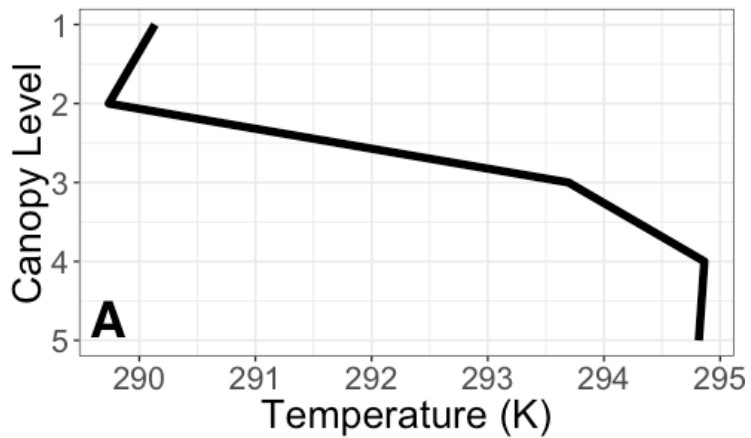

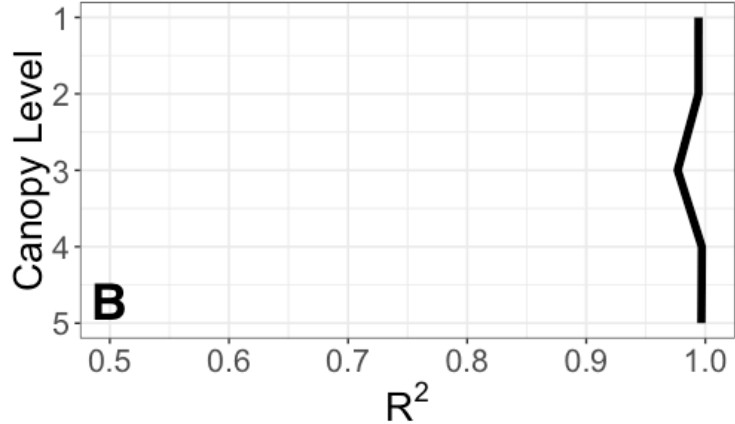

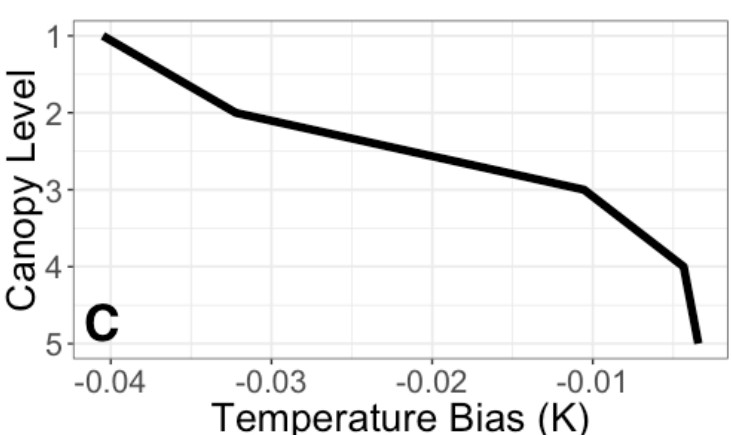





**Figure 4. Vertical profile average temperature performance of the surrogate model for 2012. Panel A shows the vertical average surrogate model leaf-level temperature (Kelvin). Panel B shows the surrogate model R² against the full model. Panel C shows the leaf level temperature bias (K) of the surrogate model compared to the full model.**

### 3.1.2 Photosynthetically Active Radiation

The annual canopy average leaf-level PAR for 2012 is shown spatially in Figure 5a.

In Figure 5, canopy temperature averages are calculated using the same method as for canopy temperature. Annual average PAR varies from ~200 µmol m$^{-2}$ s$^{-1}$ to greater than 600 µmol m$^{-2}$ s$^{-1}$. This spatial variability is a function of both PAR incident on the top of the canopy (largely related to cloud cover and solar angle) and the canopy LAI. Leaf level PAR in the surrogate model varies linearly with incident PAR, and decreases exponentially with LAI. Additionally, the reduction under high LAI is exacerbated due to a higher fraction of shaded leaves in high LAI canopies, which have substantially lower average PAR. The highest leaf-level PAR values are generally located in arid regions, where LAI and the number of cloudy days are quite low. The lowest values are located in the equatorial tropical rainforests and the northern boreal forests. The low values in the rainforest are coincident with the highest LAI values globally, leading to very strong shading effects below the first canopy layer. The northern boreal forests are low in part due to relatively high LAI, but also due to reduced incoming PAR in the winter months when the solar angle is low.

The annual average performance of the surrogate leaf-level PAR relative to the full model is shown in Figures 5b and 5c. The temporal R² over a full year in Figure 5b, is generally quite high indicating that the surrogate formulation captures the majority of the temporal variability inherent in the full model. The R² values range from 0.92





to 1.0. The highest $R^2$ values are in regions with low LAI, where the effects of

shading and other canopy physical processes are greatly reduced. The worst model

$R^2$ performance is over regions with the highest LAI, where the elevated importance

of shading and resulting complexity in the PAR calculation is more challenging for

the simplified representation of the surrogate model. However, this poor

performance still has a quite high $R^2$, with the lowest value of 0.92. The annual

average model biases are generally within ±40 μmol m$^{-2}$ s$^{-1}$, with a few more

extreme values reaching ±200 μmol m$^{-2}$ s$^{-1}$. The surrogate model is broadly biased

high over regions with lower LAI and slightly low over regions with high LAI. In a

relative sense, these changes are nearly all within 10-15%, with a maximum

normalized mean bias of 0.4.


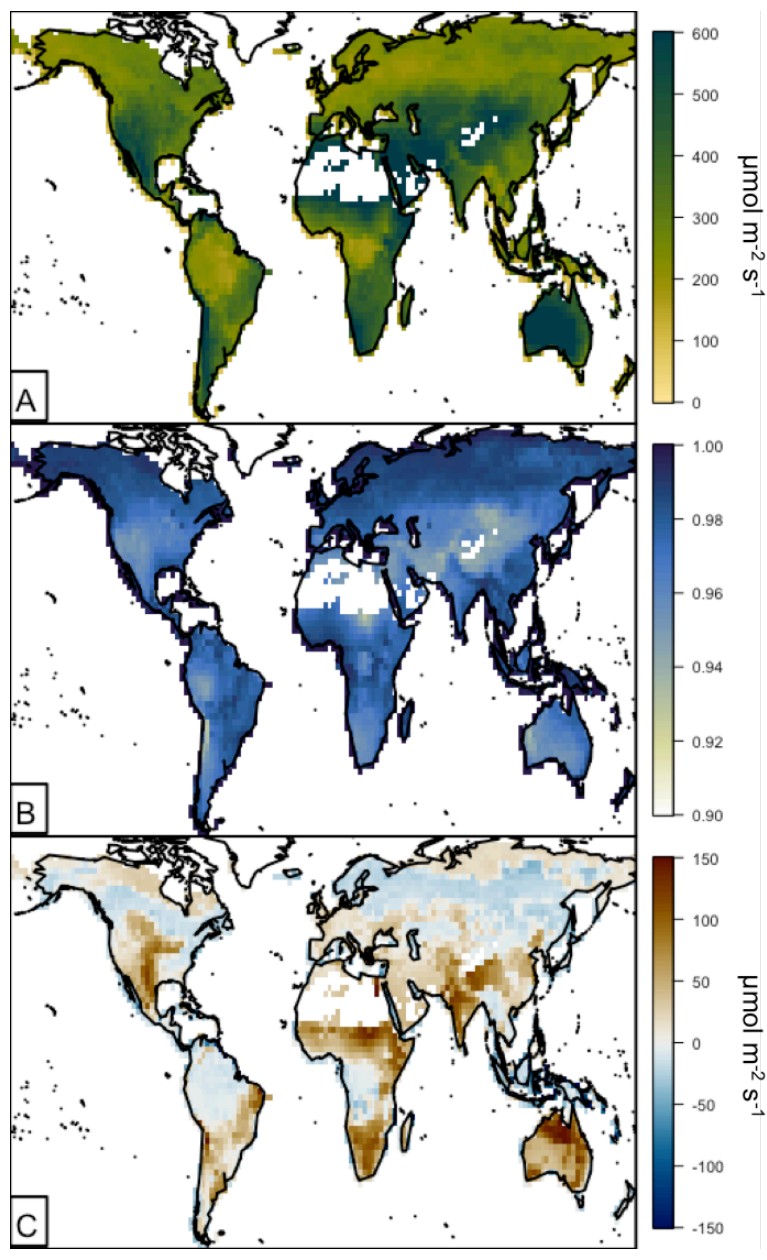

**Figure 5. Canopy spatial PAR surrogate model performance for 2012. Panels are as follows: A) Annual average surrogate model leaf level PAR (µmol m$^{-2}$ s$^{-1}$), B) R$^2$ between the leaf level PAR simulated using the surrogate and the full model, C) Annual average leaf level PAR bias (surrogate-full model, µmol m$^{-2}$ s$^{-1}$).**



The average vertical distribution of leaf-level PAR throughout the canopy and the

associated surrogate model performance are shown in Figure 6. To explore the

additional dependence on LAI, the quantities shown are separated into three LAI

ranges. These are as follows: a low range with LAI less than 0.5, a midrange with LAI

between 0.5 and 5, and a high LAI range containing canopies with total LAI greater

than 5. The low LAI range represents ~40% of all vegation throughout the year, the

middle range represents nearly 60%, and the high LAI range contains only a small

fraction of all vegetation (~1%).

The average distribution of PAR across canopy levels is shown in Figure 6a.  As LAI

increases, there is a substantial reduction in leaf level PAR deeper into the vegetated

canopy. This is particularly obvious with the high LAI range, consistent with

substantial shading and light interception above the bottom of densely vegetated

canopies. On the other hand, the variability throughout the low LAI canopies is quite

small. This LAI dependence explains in part the relatively low canopy average leaf-

level PAR throughout the tropical forests in Figure 5a. The variability in the PAR at

the top canopy layer (Canopy Level 1 in Figure 6a) stems from two major sources.

The first is simply the spatial distribution of these LAI ranges in relationship to the

annual average incident PAR to the canopy top. Very high LAI values occur primarily

over the tropics, where sunlight is consistently high throughout the year, and the

seasonal effects of changing solar angles is small. The opposite is true for many of

the regions with smaller LAI values, which are distributed more evenly across the

globe. A second order effect in the MEGAN3 canopy model is that of in-layer

attenuation of light and shading throughout the canopy, where leaves in a given



layer may intercept light and shade leaves lower within that same layer. This has the effect of reducing the layer average leaf-level PAR as a function of leaf geometries

and LAI, and explains why the highest canopy layer average leaf-level PAR is not the same as the average PAR incident on the top of the canopy.

Figures 6b and 6c summarize the statistical performance of the surrogate model vertically through the canopy in terms of the $R^2$ and the mean bias, respectively. Overall, the surrogate model reproduces the PAR variability as compared to the full

parent model well. For both the low and middle LAI ranges (LAI less than 5), all $R^2$ values are greater than ~0.9. The only substantially lower $R^2$ are from the lower canopy in high-LAI regions, where PAR is generally quite small (see Figure 6a). While the surrogate model struggles to capture this lower canopy variability, the ultimate influence on the canopy-scale bias is generally low.

The vertical distribution of that bias is shown in Figure 6c. Broadly, the absolute PAR bias is low (less than 5% on a relative scale) and decreases throughout the canopy. All biases are positive except for the top canopy layer for high LAI range canopies, and this poor fit is likely related to the limited representation of high LAI regions in the full dataset (only ~1% of all vegetated area), and isn't present if a

lower cutoff for high LAI ranges is used (e.g. LAI > 3). The decreasing magnitude throughout the canopy is largely related to the decreasing overall leaf-level PAR (see Figure 6a). It important to note that the bias terms are all sensitive to the choice of LAI bin ranges, and the variability in bias at each level can be quite large (e.g. above 50 µmol m$^{-2}$ s$^{-1}$ in the top canopy layer). For both the high and middle LAI ranges,





the absolute magnitude of the PAR bias decreases throughout the canopy, and the

bias remains relatively constant for low LAI range vegetation. On a relative scale,

these biases are all quite small with a normalized mean bias usually less than 5%.

The exception to this is the lowest layer of the high LAI range canopies. In this low

canopy layer the magnitude of the bias is quite low, as is the total magnitude of leaf-

level PAR, the resulting difference between small numbers leads to a relatively large

normalized mean bias of  ~0.3.



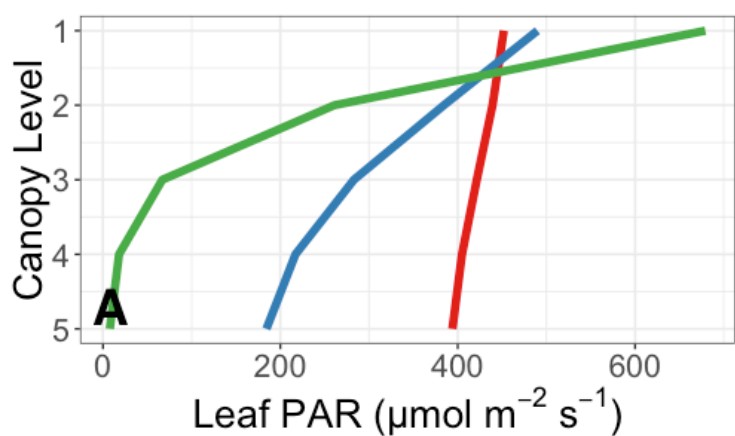

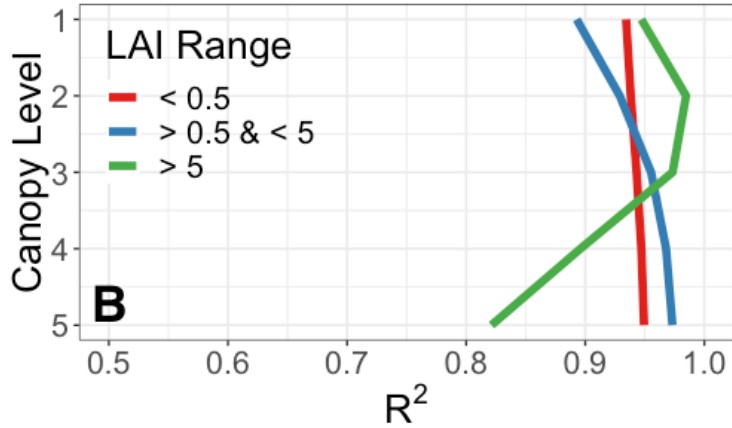

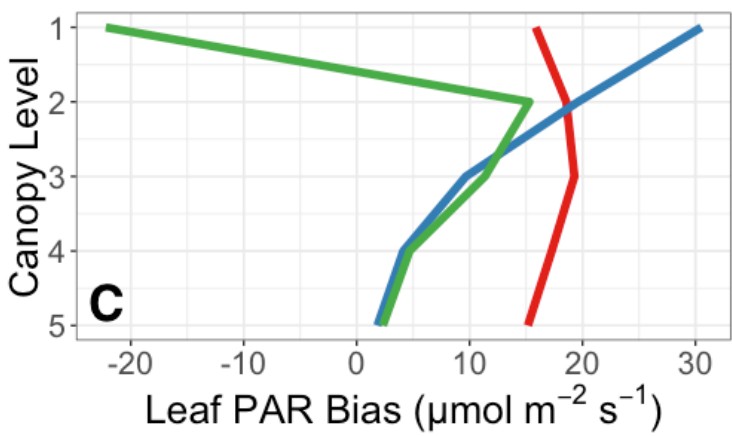





**Figure 6. Vertical profile average PAR surrogate model performance as a function of LAI for 2012. Panel A shows the vertical average surrogate model leaf-level PAR (μmol m$^{-2}$ s$^{-1}$) for low LAI (red), mid-range LAI (blue) and high LAI canopies. Panel B shows the surrogate model R$^2$ against the full model. Panel C shows the leaf level PAR bias (μmol m$^{-2}$ s$^{-1}$) of the surrogate model compared to the full model. Level 1 is the top of the canopy.**

An essential function of canopy models used in CTMs is to calculate the amount of light that falls on already light-saturated leaf surfaces. This regulates the effect of a change in PAR incident on the canopy on various biological and physical processes (e.g. biogenic emissions). We estimate the fraction of leaves that are light-saturated using the $\gamma_{PAR}$ formulation from the MEGAN algorithm (Guenther et al 2006, 2012). This variable aims to capture the amount of light saturation on a given leaf, and ranges from 0 to 1, with higher values corresponding to more saturated leaves. To explore light saturation, we examine cases where the $\gamma_{PAR}$ value is greater than 0.9. A scatterplot of the annual average fraction of leaves that are light-saturated ($\gamma_{PAR} \geq$ 0.9) per model gridbox for both the full model and the surrogate model throughout the canopy is shown in Figure 7. The surrogate model reproduces the full model fraction of light-saturated leaves well, generally to within ~5%, with a median bias of -2%.



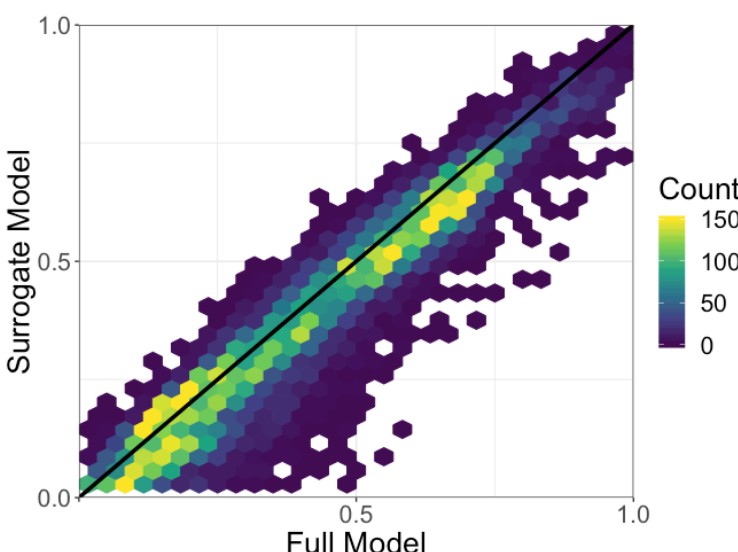

**Figure 7. The annual grid box average fraction of light saturated leaves as simulated by the full and surrogate models throughout the canopy for the year 2012. The colorbar represents the number of observations in a given hex. The 1:1 line is shown in black.**

Ultimately, this assessment demonstrates that the surrogate model reproduces the

parent MEGAN3 canopy model well for both leaf temperature and leaf-level PAR.

The exponential relationship between leaf-level PAR and canopy incident PAR and

the linear relationship between leaf temperature and near-surface air temperature

captures the majority of the information inherent in the parent model. Some higher

order variability in absolute magnitude of the variables is missing from this

surrogate model, however the biases are generally all within ~10%.

**4. Chemical Transport Model Description**

We evaluate the impact of the canopy model parameterization on atmospheric

composition using the GEOS-Chem v12.3.0 chemical transport model (www.geos-





chem.org). GEOS-Chem is a computational model for simulating atmospheric

chemistry, including a detailed $HO_x$-$NO_x$-$BrO_x$ tropospheric chemical mechanism

(Bey et al., 2001; Mao et al., 2013; Travis et al., 2016). We drive GEOS-Chem with

MERRA-2 Meteorology at 2°x2.5° spatial resolution, with 47 vertical layers (Gelaro

et al., 2017). The timesteps for convection and chemistry are 10 and 20 minutes,

respectively.  Identically to the canopy model input data, we use LAI values from the

MODIS-Terra MOD15A2 product (Myneni et al., 2002, 2007), and plant functional

types from the Olson 2001 dataset (Olson et al., 2001). Fire emissions are from the

Global Fire Emissions Database v4 (GFED4, Giglio et al., 2013), and global

anthropogenic emissions are from the Community Emissions Data System inventory

(CEDS, Hoesly et al., 2018). Regional emissions over the United States, Africa, and

Asia are from the NEI 2011 (Travis et al., 2016), DICE-Africa (Marais and

Wiedinmyer, 2016), and MIX (Li et al., 2017) emissions inventories, respectively.

Soil $NO_x$ emissions are calculated following Hudman et al. (2012). Simulations are

shown for the years 2012 and 2013, with the first year discarded for spin up when

considering gas-phase chemical impacts.

### 4.1 MEGAN Emissions

The biogenic emissions scheme in GEOS-Chem v12.3.0, MEGAN2.1, is based on

Guenther et al. (2006, 2012) and Millet et al. (2010). The emissions of a given

compound are calculated from base canopy-level emission factors multiplied by

"activity factors" representing standard processes that govern biogenic emissions

(temperature, PAR, light-dependence, etc.), and "stress factors" modeling the effect

of vegetative stress (heat, drought, etc.) on biogenic emissions. Each of these activity



and stress factors vary with the environmental state. The base emission factor itself

varies with vegetation type, and these activity factors respond to leaf temperature,

leaf-level PAR, leaf age, leaf area index, soil moisture, and atmospheric $CO_2$

concentrations. The base emission factors used in this work are consistent with

those used in GEOS-Chem v12.3.0; an example for isoprene is shown in Figure 8. The

emission factors are highest in forested regions, and lowest over areas with little

vegetation (e.g. deserts). These emission factors are regridded from the original

resolution of 0.25°x0.3125° to match the GEOS-Chem resolution of 2°x2.5°.

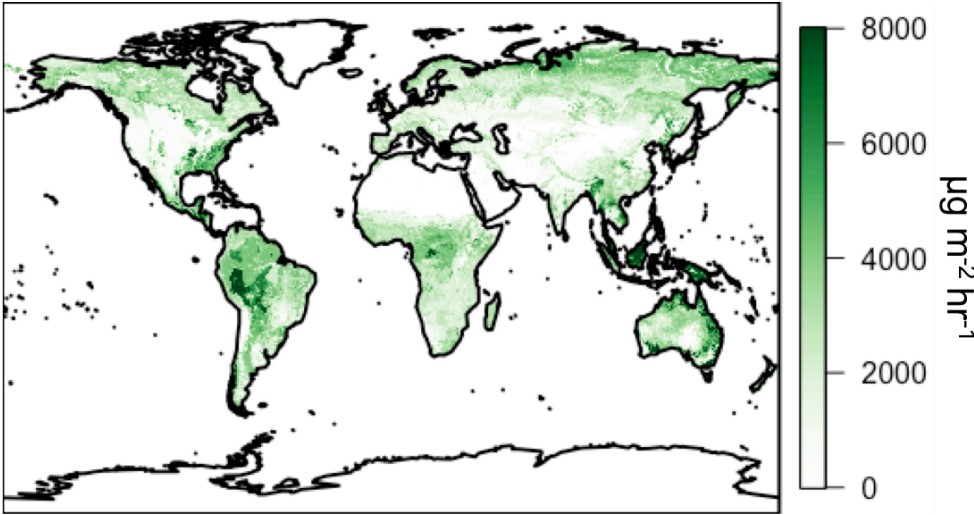

**Figure 8. Base Isoprene emission factors used in this work.**

As GEOS-Chem v12.3.0 has no representation of plant canopy physics, the LAI,

temperature, and PAR activity factors are all re-parameterized following Guenther

et al. (2006) in the standard model. In this parameterization, leaf temperature is

assumed to equal air temperature in the calculation of the temperature activity

factor. The LAI and PAR activity factors are calculated in an approach known as the

Parameterized Canopy Environment Emission Activity (PCEEA) approach that does





not include any description of the vertical distribution of vegetation, and only

includes response to the LAI, PAR incident to the top of the canopy, and the solar

zenith angle.

We modify the MEGAN implementation in GEOS-Chem to allow for the

representation of canopy physics decribed in Section 3. In order to properly scale all

emission factors to the plant canopy using a canopy model, a normalization factor

must be applied at a set of standard environmental and ecological conditions

(Guenther et al., 2006, 2012). This normalization factor varies depending on the

choice of those standard conditions and the canopy model used. In MEGAN2.1 these

standard conditions are: LAI of 5, current air temperature of 303K, current incident

PAR at the canopy top of 1500 μmol m$^{-2}$ s$^{-1}$, and a 10%/80%/10% split of growing,

mature, and senescent leaves (Guenther et al., 2012; Kaiser et al., 2018). We

calculate other necessary standard conditions, specifically the 24-hour average air

temperature and PAR, from the meteorological fields conditional on locations that

meet the previously described standard conditions. In situations where these

previous standard conditions (e.g. Current Temperature = 298.5K, Current PAR =

1500 μmol m$^{-2}$ s$^{-1}$) are met to within 10%, we calculate the 24-hour average prior

meteorological conditions from the reanalysis fields, and use the mean of those

calculations as the standard 24-hour average conditions. The resulting standard

conditions for 24-hour average temperature and 24-hour average PAR are 298.5 K

and 740 μmol m$^{-2}$ s$^{-1}$, respectively. These standard conditions result in a

normalization factor of 0.21 using the surrogate canopy model surrogate developed

in this work. The value of 0.21 is lower than those used in implementations of





previous MEGAN model versions in other models such as CLM (0.3) and WRF-Chem (0.57) (Guenther et al., 2012). It is important to note that small deviations from these standard conditions ($\pm 3$ K or $\pm 20$ µmol m$^{-2}$ s$^{-1}$) can lead to changes in the

normalization factor of nearly 20%. Since this normalization factor is applied consistently to all emissions globally at all times, it linearly modulates all biogenic emissions. As such, the total emissions calculated by the MEGAN2.1 emissions framework are highly sensitive to the parameter choices made in this normalization processes.

In GEOS-Chem v12.3.0 we update the activity factors associated with PAR, LAI, and temperature as well as the normalization to take advantage of our new canopy surrogate model. This enables a full implementation of the most recent MEGAN3 emission activity algorithm in the GEOS-Chem model. In the PCEEA implementation of MEGAN in the base version of GEOS-Chem, activity factors are calculated

separately for PAR ($\gamma_P$), LAI ($\gamma_{LAI}$), and temperature ($\gamma_T$), and then multiplied together following Guenther et al. (2006):

(5) $\gamma_{PCEEA} = \gamma_{LAI}\gamma_T\gamma_P$

Following MEGAN3 (https://bai.ess.uci.edu/megan), we implement PAR and temperature activity factors that are calculated jointly per canopy level and summed

together weighted by the vertical canopy biomass distribution. In this work, as in previous non-PCEEA versions of the MEGAN framework (Guenther et al., 2006, 2012), the effect of LAI is calculated through direct multiplication of the emission





factor by LAI as opposed to an activity factor formulation, along with a canopy normalization factor ($C_{CE}$):

(6) $\gamma_{Canopy} = C_{CE} LAI \gamma_{TP}$

(7) $\gamma_{TP} = \sum_{l=1}^{5} w_l \gamma_P \gamma_T$

These activity factors for PAR, LAI, and temperature are the same as those in Guenther et al. (2012), as averages throughout the canopy weighted by the biomass fraction within a given canopy layer ($w_l$). There is an additional canopy depth

emission activity response applied to the light dependent activity factors which is intended to model the variability of emissions throughout the canopy (e.g. Harley et al., 1996). This canopy depth activity factor is a multiplicative factor that varies linearly as a function of LAI and canopy depth, with a value between 0 and 1.3. For clarity, we refer to the MEGAN emissions implementation in GEOS-Chem using the

$\gamma_{PCEEA}$ activity factors as "MEGAN$_{PCEEA}$" and those using the $\gamma_{Canopy}$ activity factors as "MEGAN$_{Canopy}$". We use the canopy physics surrogate model described in Section 3 to calculate the leaf temperature and PAR in the MEGAN$_{Canopy}$ implementation.

Though stress factors in the MEGAN framework allow for the additional capability to evaluate the impact of vegetative stress processes on emissions (e.g. Geron et al.,

2016), we do not enable those processes in this study. The other activity factors (leaf age, soil moisture, and $CO_2$ inhibition) are the same in both the MEGAN$_{Canopy}$ and MEGAN$_{PCEEA}$.



### 4.2 Dry Deposition

Dry Deposition in GEOS-Chem v12.3.0 is calculated through a resistor-in-series

approach based on the Wesely (1989) parameterization, originally described and

implemented in Wang et al. (1998). In this approach, the dry depositional flux of gas

phase species is calculated as the surface concentration of that gas multiplied by a

transfer velocity known as the "dry deposition velocity". A recent assessment of the

dry deposition velocity parameterization in GEOS-Chem found that biases in

simulated dry deposition velocities are in general quite low, though there is

evidence that missing key processes may be responsible for missing variability in

the simulation (Silva and Heald, 2018).

Prior to this work, canopy effects were not directly considered in GEOS-Chem dry

deposition, and only approximated in bulk using a polynomial decomposition

scheme (Wang et al., 1998) that jointly calculated both a multiplicative factor $(1 + b/PAR_{leaf}$, b = 50 μmol m$^{-2}$ s$^{-1}$) to the stomatal resistance from Baldocchi et al. (1987)

based on leaf-level PAR and a normalization of the stomatal resistance by LAI. Here,

we replace the polynomial decomposition scheme and use the leaf-level PAR

calculations from the canopy surrogate to directly calculate the multiplicative factor,

and then explicity normalize by LAI. The LAI normalization in the original

polynomial decomposition calculates values that are a factor of ~1.7 higher than

those calculated through direct normalization when using the surrogate model. To

maintain the same magnitude of the simulated dry deposition velocities as in the

standard model, which are generally unbiased (Silva and Heald, 2018), we scale the

stomatal resistance by a factor of 0.6.



## 5. Surrogate Model Integration into GEOS-Chem

Implementing the updated canopy surrogate in a global model directly impacts the surface-atmosphere exchanges processes of biogenic emissions and dry deposition, which together influence the chemical composition of the atmosphere. In this

section we outline the changes to both surface processes, focusing on isoprene emissions and ozone dry deposition, followed by the changes to surface level ozone concentrations in the GEOS-Chem model.

The impact of the canopy model on isoprene emissions in 2012 is summarized in Figure 9. The annual average isoprene emissions using the MEGAN$_{Canopy}$ emissions

implementation are shown in Figure 9a, with the highest emissions in the tropics and subtropics, as well as the Southeast United States. Though not distinct in Figure 9, the boreal forests are a substantial emitter of biogenic species during the summer months. The relatively small emissions from this region during the winter months reduce the prominence of these emissions on the annual average. The global annual

total of isoprene emitted in 2012 from the MEGAN$_{Canopy}$ configuration is ~350 Tg C yr$^{-1}$.

The annual average differences in the simulated isoprene emissions following implementation of the surrogate canopy model (MEGAN$_{Canopy}$ − MEGAN$_{PCEEA}$) are shown in Figure 9b. In general, emissions decrease over forested regions and

increase over non-forested (grasses, crops, and shrubland) areas. The highest absolute changes are the decrease in the Equatorial Amazon and the increase in Northern Australia. On a relative scale, the forested and non-forested differences are





more apparent. This relative change (MEGAN$_{Canopy}$/MEGAN$_{PCEEA}$) is shown in Figure

9c. While there are relatively modest decreases in tropical and boreal forests, the

emissions increase in heavily cropped Indian Subcontinent and sub-Saharan Africa

show the largest relative change. Though the spatial variability in relative difference

is substantial, the annual global isoprene emissions from the canopy model are

within 5% of the original model version (~340 Tg C yr$^{-1}$). These results are

consistent with those from Guenther et al. (2006) who found that the global total

biases in isoprene emissions were low, but spatial variability was large, when using

a parameterized approach (MEGAN$_{PCEEA}$) over a direct canopy model

implementation in the MEGAN framework (as in the surrogate model application,

MEGAN$_{Canopy}$).





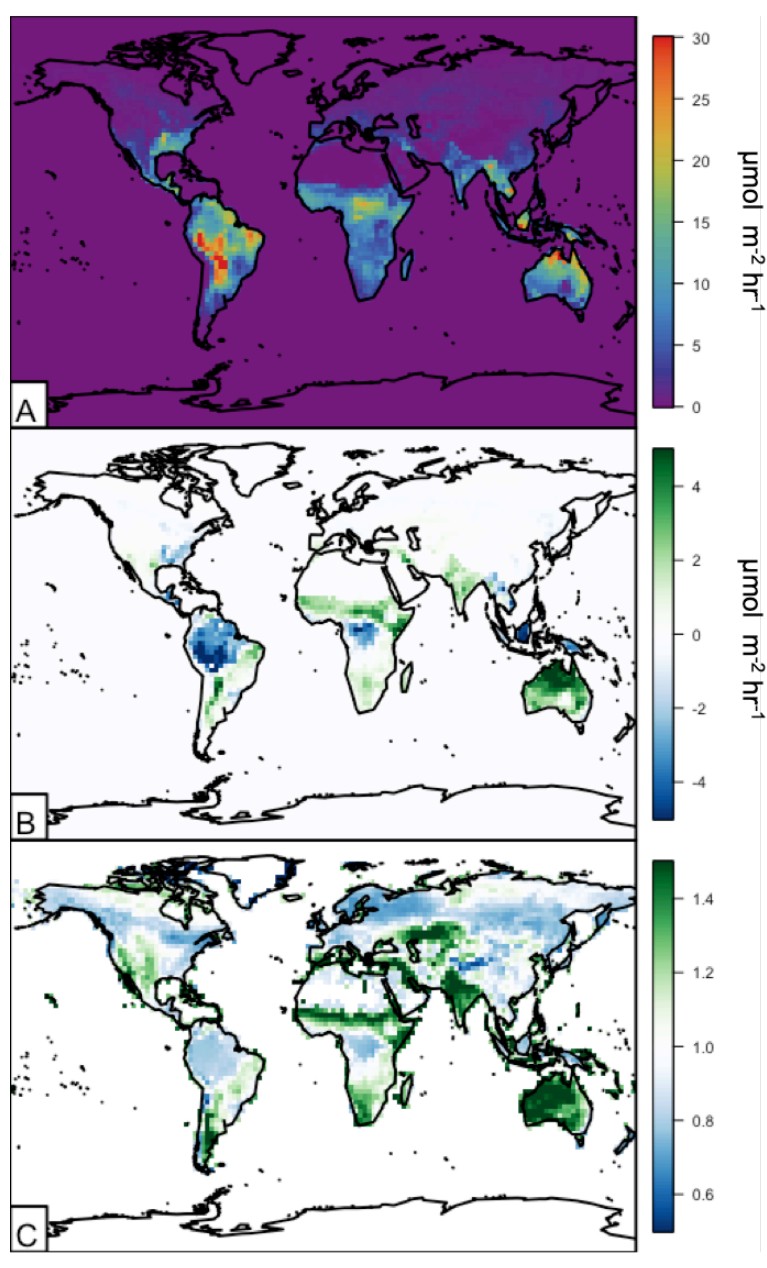

**Figure 9. Annual average (2012) isoprene emissions simulated in GEOS-Chem driven by the surrogate model canopy physics (MEGAN$_{Canopy}$). Panel A shows the annual average emissions. Panel B shows the difference between the surrogate model and base version of simulated emissions. Panel C shows the relative difference between the surrogate model and base version of simulated emissions (surrogate/base model).**



It is not possible to directly parse the individual process contributions to the total

emissions changes due to the different coupled treatments of the influence of

temperature, PAR, and canopy structure on biogenic emissions through the activity

factors in both the MEGAN$_{Canopy}$ and the MEGAN$_{PCEEA}$ configurations. However, a

comparison of the isoprene differences between the two simulations against LAI,

leaf-level PAR, and leaf temperature (Figure 10) indicates that the changes are most

strongly driven by the leaf-level PAR and LAI effects. The isoprene emissions

changes are directly proportional to leaf-level PAR, inversely proportional to LAI,

and show no substantial relationship to leaf temperature. The forested and non-

forested differences in Figure 9 can be explained further from the correlations

shown in Figure 10. The forested areas with the largest decreases in isoprene

emissions tend to have high LAI values and lower canopy average leaf PAR, whereas

the opposite is true for the non-forest locations. The relationships in Figure 10

support the interpretation that the leaf-level PAR and LAI effects are the largest

drivers of change in biogenic isoprene emissions between the two model versions.

Overall, these results indicate that the representation of canopy radiative physics is

more important than thermodynamically resolving the difference between air and

leaf temperature for simulating biogenic emissions in the MEGAN framework.



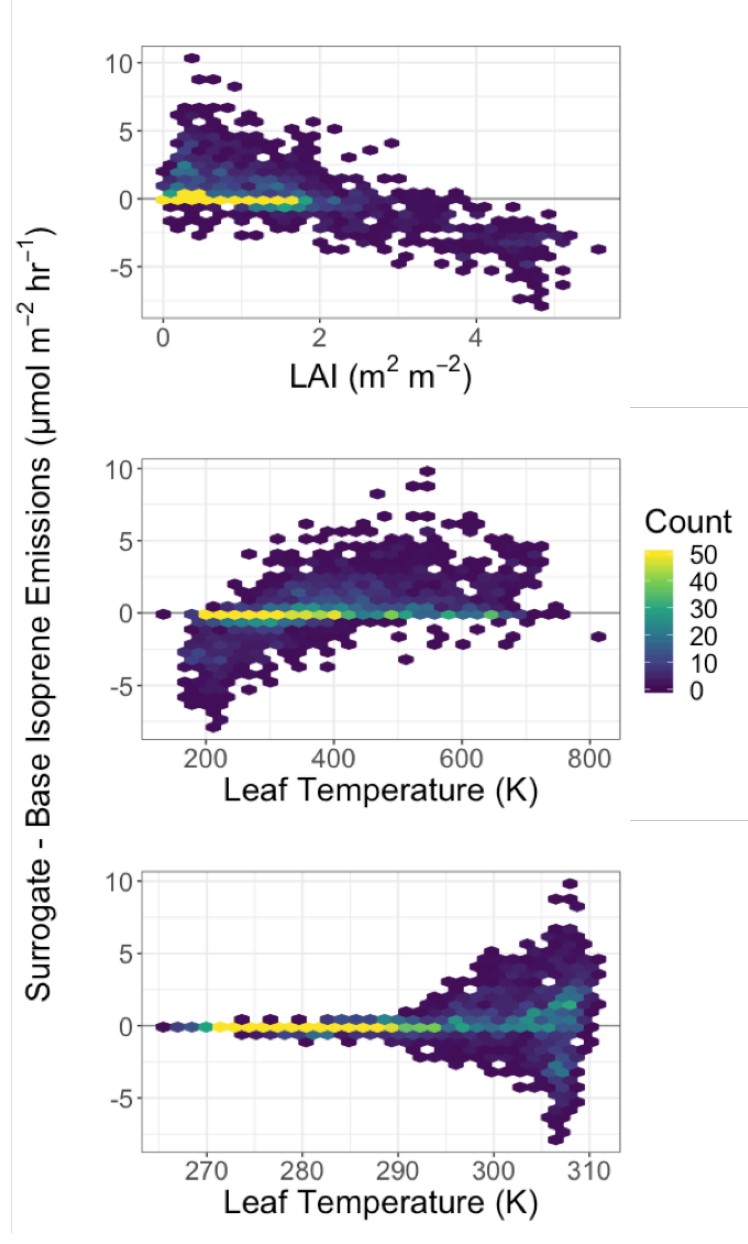

**Figure 10. Difference in annual average isoprene emissions between the surrogate canopy model (MEGAN_Canopy) and the base simulation (MEGAN_PCEEA) (atoms C cm$^{-2}$ s$^{-1}$, see Figure 9B) as a function of LAI, Leaf Level PAR (μmol m$^{-2}$ s$^{-1}$), and Leaf Temperature (K). Gridboxes dominated by water were filtered removed from these figures. The colorbar represents the number of observations in a given hex.**






There are few spatial constraints on isoprene emissions that can act as independent

validation data for the new model framework. However, recent work over the

southeast United States (Kaiser et al., 2018; Travis et al., 2016; Yu et al., 2016)

indicates that the base version of GEOS-Chem used here (v12.3.0), which uses

MEGAN$_{PCEEA}$, overestimates isoprene emissions by 15-40%. The MEGAN$_{Canopy}$

configuration reduces isoprene emissions in most locations in the Southeast United

States by ∼10%, and locally leads to reductions as large as ∼20%, bringing the

model into better agreement with these observational constraints.

The MEGAN emissions framework calculates the emissions of other non-isoprene

biogenic species as well, including monoterpenes. The influence of the canopy

surrogate model on monoterpene emissions is shown in Figure 11. The annual total

monoterpene emissions in 2012 from MEGAN$_{Canopy}$ are ∼95 Tg C yr$^{-1}$. These

emissions are shown in Figure 11a and are highest over the densely vegetated

regions of the world, in particular the tropics. Similar to isoprene emissions,

monoterpene emissions in the northern latitude forests peak during summer

months. The implementation of the canopy surrogate model reduces global annual

total monoterpene emissions by approximately 20%. The annual average absolute

and relative changes to monoterpene emissions due to the canopy surrogate model

(MEGAN$_{Canopy}$ − MEGAN$_{PCEEA}$) are shown in Figures 11b and 11c, respectively.

Simulated monoterpene emissions differ from isoprene emissions in that

monoterpene emissions are more sensitive to temperature, with an additional

influence of a light independent emission factor that varies with leaf temperature

(Guenther et al., 2012). There is a fairly constant 20-30% decrease across regions





with lower LAI values, including the African savannahs and the Indian subcontinent. The highest absolute changes are in transitional areas near high LAI forests, with warmer temperatures (the tropics and subtropics). The high LAI areas of the tropical and northern forests show smaller decreases of ~5%. These changes, while

substantial, are well within the stated uncertainty ranges in monoterpene emissions of the MEGAN model (300-400%, Guenther et al., 2012).



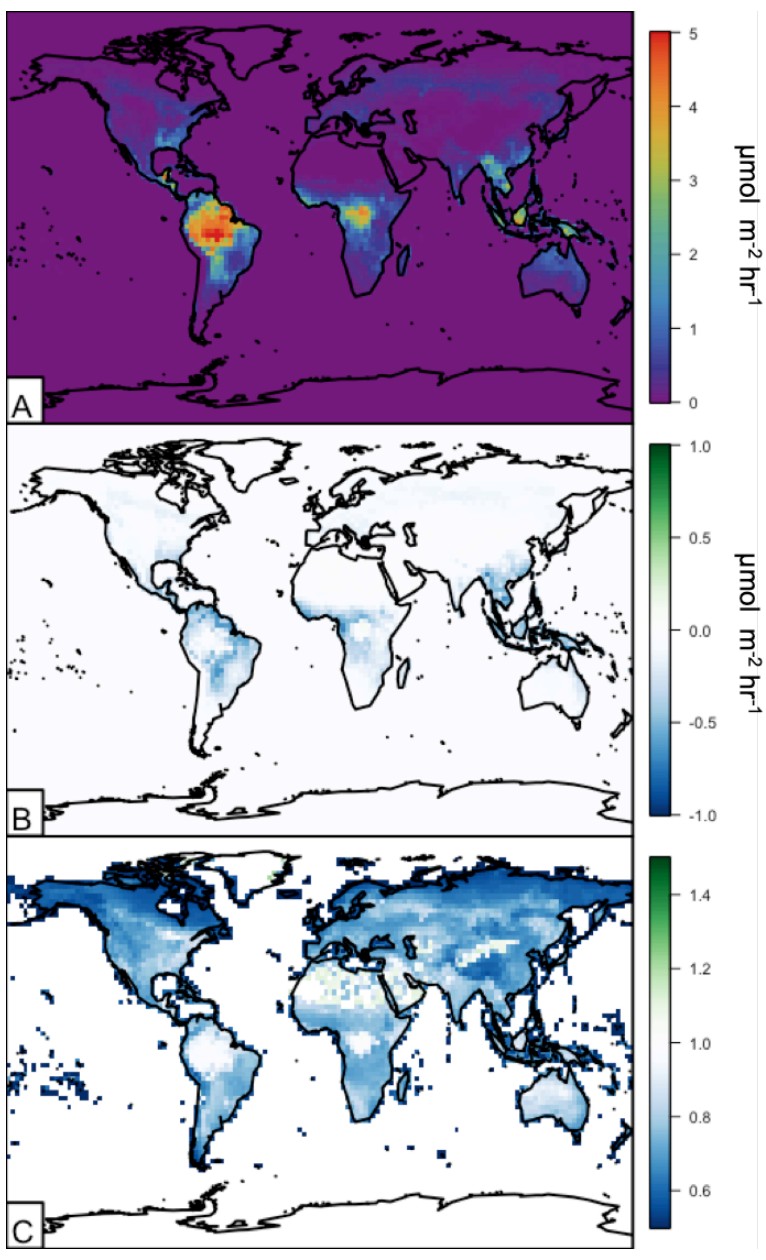

**Figure 11. Annual average (2012) monoterpene emissions simulated in GEOS-Chem driven by the surrogate model canopy physics. Panel A shows the annual average emissions. Panel B shows the difference between the surrogate model and base version of simulated emissions. Panel C shows the relative difference between the surrogate model and base version of simulated emissions (surrogate/base model).**






Changes in simulated ozone dry deposition velocities in 2012 are summarized in

Figure 12. Figure 12a shows the annual average spatial distribution of ozone dry

deposition velocities. The values vary from less than 0.1 cm s$^{-1}$ over the global

oceans, to above 0.5 cm s$^{-1}$ in densely vegetated regions like the tropical rainforests.

The impact of the updated canopy model on ozone dry deposition velocities is in

general quite small, with an average change of near zero (~0.004 cm s$^{-1}$). The annual

average relative change is shown in Figure 12b, and the absolute difference in

Figure 12c, both in relation to the base version of GEOS-Chem v12.3.0. These

changes are nearly all within ±5%, or ±0.01 cm s$^{-1,}$ with a maximum change of 15%,

(0.04 cm s$^{-1}$). Relative changes track most strongly with broadleaf and coniferous

forested areas, consistent with those regions being most sensitive to stomatal

deposition (Silva and Heald, 2018).

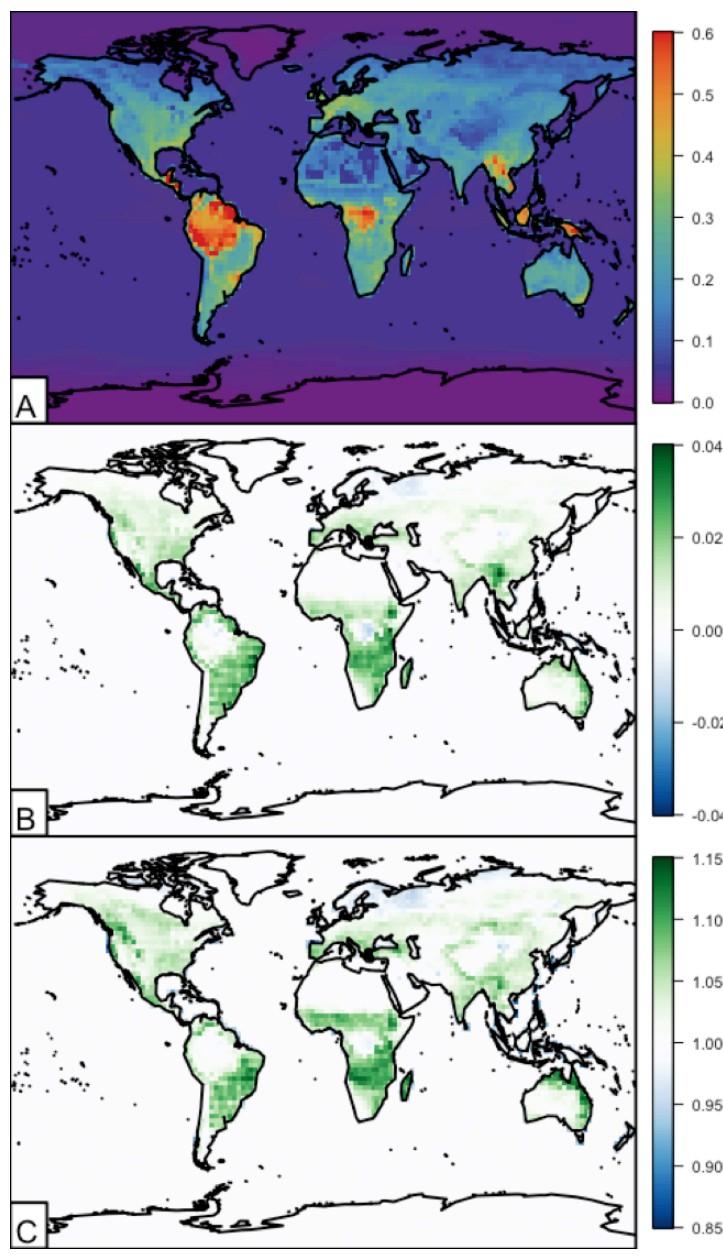

**Figure 12.** Annual average (2012) ozone dry deposition velocities simulated in GEOS-Chem when driven by the surrogate model canopy physics. Panel A shows the annual average dry deposition velocities (cm s$^{-1}$). Panel B shows the difference between the surrogate model and base version of simulated dry deposition velocities (cm s$^{-1}$). Panel C shows the relative difference between the surrogate model and base version of simulated dry deposition velocities (surrogate/base model).




The small overall changes to surface-atmosphere exchange processes associated with the updated canopy scheme produce only a modest impact on simulated atmospheric composition. We describe the changes to surface ozone here, as an

illustrative example.

The annual average spatial difference in surface ozone between a simulation using the canopy physics described here and the base version of GEOS-Chem is shown in Figure 13. These changes are all generally quite small; all are within 10% of the base simulated annual averages. The changes are generally within ±1 ppbv, with the

largest absolute changes over regions with the largest changes in isoprene emissions. The distribution of differences largely follows well-known $NO_x$-VOC ozone formation patterns. The $NO_x$ limited regions of the world, in particular the remote tropics, show an inverse relationship with isoprene emissions. This is consistent with ozone titration by isoprene in the presence of low $NO_x$. The largest

changes in ozone over the VOC limited regimes of India and China correspond directly to the changes in isoprene emissions changes, with enhanced isoprene emissions over the Indian subcontinent increasing ozone concentrations, and the decrease in isoprene emissions over China leading to a decline in ozone. The overall influence of the changes in ozone dry deposition velocity is fairly negligible. Even

the regions where the dry deposition velocity change is the largest (e.g. the Amazon) are dominated by the shift in isoprene emissions.

In total, the changes in surface ozone concentrations slightly ameliorate known biases. There is a persistent high bias (~10 ppbv) across chemical transport models





in simulating surface ozone concentrations over the continental mid-latitudes

(Travis et al., 2016). The addition of the new canopy physics parameterization very

modestly reduces this bias by about ~1ppbv, driving simulated ozone closer to

observations.

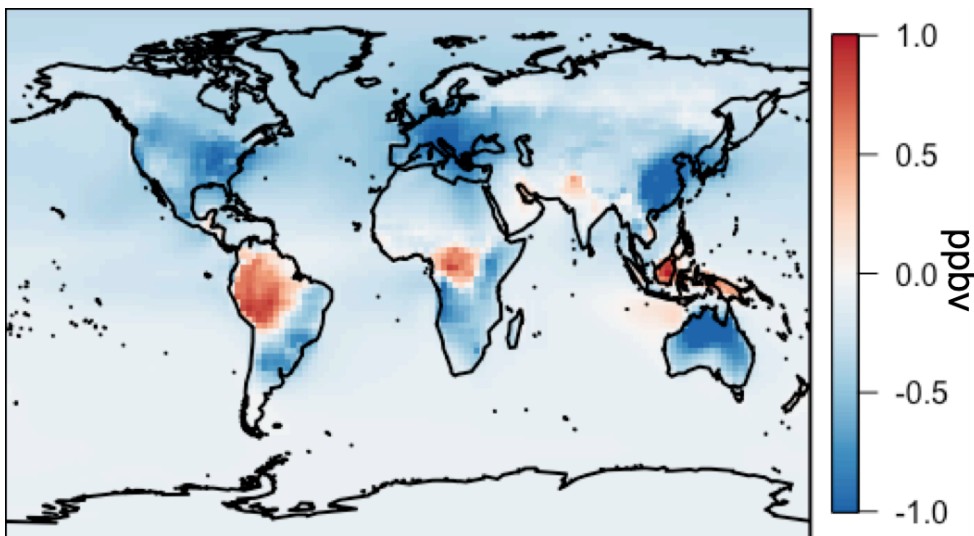

**Figure 13. Annual spatial average surface ozone difference (ppbv) between the updated model version**
**with surrogate canopy physics and the base version of GEOS-Chem (surrogate-base).**

## 6. Implementation of MEGAN3 Emission Factors

In addition to improved process representation, the canopy surrogate model

presented here allows for the direct application of new emission factors generated

using the MEGAN3 Emission Factor Processor (https://bai.ess.uci.edu/megan),

which allows users to generate emission factors from various input datasets. While

the focus of this work is on the impact of representing canopy physics, we include

here a description of this full implementation of MEGAN3 in the GEOS-Chem model

for completeness. We calculate landscape-average emission factor distributions



using the global growth form and ecotype distributions, the emission type

speciation, and the leaf level emission factor database available from the MEGAN3

Emission Factor Preprocessor. All MEGAN3 Emissions Factor Preprocessor options

we kept at their defauly values (i.e. confidence rating J = 0, and 20 total species

classes).  The landcover and emissions data are the same as that used for MEGAN2.1

except for some updates for the contiguous US. The spatial distribution of the

MEGAN3 activity factors is shown in Figure 14. It is important to note that these

new emission factors are input at the leaf level with units on a per LAI basis, as

opposed to the canopy-scale factors used in previous versions of MEGAN (applied

earlier in this manuscript), which makes direct comparisons of emission factor

magnitudes infeasible. This canopy to leaf level change ultimately has the

consequence of removing the need for the normalization factor in the activity factor

calculation (see Section 4.1). When these emission factors are scaled to the same

units as in MEGAN2.1 using the MODIS LAI product used in this work, the resulting

emission factors are relatively similar (within ±75%) though the MEGAN3 emission

factors are lower than those used with MEGAN2.1 in GEOS-Chem. Generally, more

than half of all changes are within 1000 $\mu$mol m$^{-2}$ hr$^{-1}$, and 90% of all emission

factors are within 3000 $\mu$mol m$^{-2}$ hr$^{-1}$. This comparison is not exact, due to the fact

that the MODIS LAI product used here is likely different from the input vegetation

files used to create the original MEGAN2.1 emission factors. Despite the differences

in absolute magnitude, the spatial pattern in emission factors in Figure 14 are very

similar to those used earlier in this work (Figure 8), with a spatial $R^2$ of 0.91.



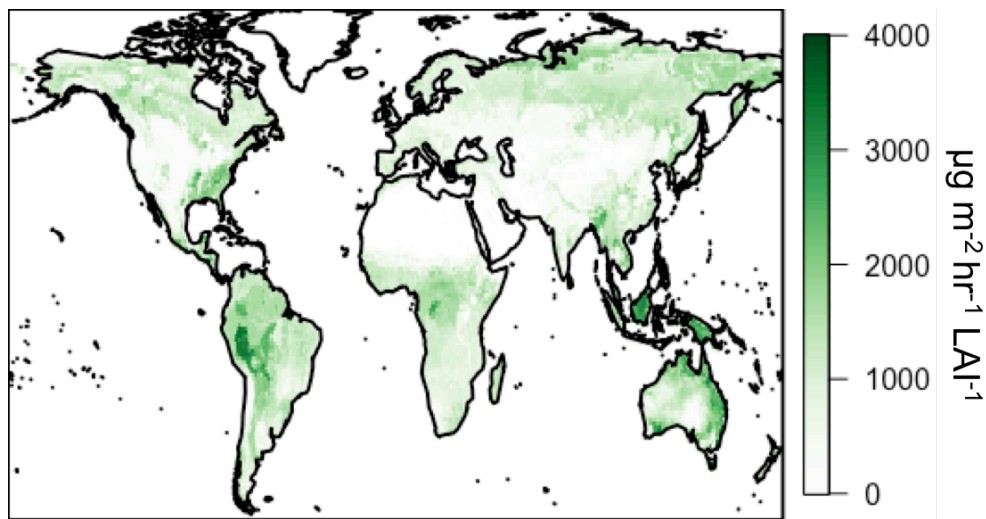

**Figure 14. MEGAN3 isoprene emission factors (μg m$^{-2}$ hr$^{-1}$ LAI$^{-1}$).**

We implement the MEGAN3 emission factors in GEOS-Chem v12.3.0 using the

canopy surrogate model activity factor formulation. For clarity, we refer to

"MEGAN3$_{Full}$" as the emissions implementation in GEOS-Chem v12.3.0 using the

MEGAN3 leaf-level emission factors and MEGAN3 activity factors with canopy

physics calculated following the canopy surrogate model described in Section 3. The

annual isoprene emissions simulated using MEGAN3$_{Full}$ are higher than using the

MEGAN2.1 canopy-scale factors in GEOS-Chem (as in both MEGAN$_{Canopy}$ and

MEGAN$_{PCEEA}$), but more in line with previous work (Guenther et al. 2012).

Specifically, annual total isoprene emissions for 2012 are ~570 Tg C yr$^{-1}$ in

MEGAN3$_{Full}$ which is a factor of 1.6 larger than those configurations discussed

earlier in this manuscript. The largest contribution to these differences is not the

differences in emission factor maps, but is instead the removal of the normalization

factor of 0.21, which additionally removes the need for the somewhat arbitrary



choice of "standard conditions" for emissions (see Section 4.1). This 570 Tg yr$^{-1}$

emissions total is much more similar to the magnitude of global emissions from

versions of MEGAN2.1 implemented outside of the GEOS-Chem model (535-578 Tg

yr$^{-1}$) given by Guenther et al. (2012). These annual average isoprene emissions

using MEGAN3$_{Full}$ are shown in Figure 15 below. In general the spatial pattern in the

emissions in Figure 15 matches those from the MEGAN$_{Canopy}$ configuration (Figure

9), with an R$^2$ of ~0.8.

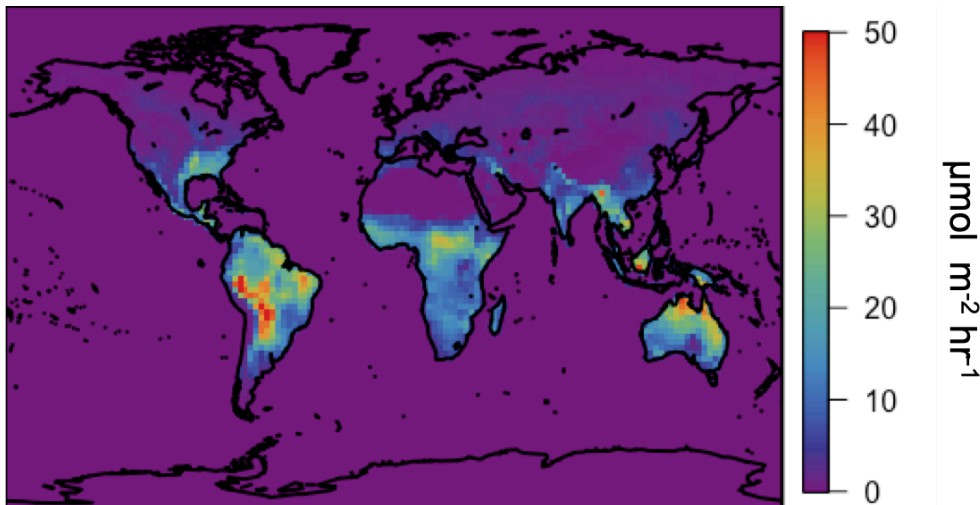

**Figure 15. Annual average (2012) isoprene emissions simulated in GEOS-Chem driven by the surrogate**
**model canopy physics and the MEGAN3 emission and activity factors.**

Since the isoprene emissions calculated using the MEGAN3$_{Full}$ algorithm are so much

larger than those used in previous versions of GEOS-Chem, they alter the

composition of the atmosphere significantly. For example, annual average surface

ozone concentrations in the Southeast U.S. increase by nearly 5 ppbv relative to the

base version of GEOS-Chem v12.3.0 (which uses MEGAN$_{PCEEA}$), exacerbating the

existing model bias further (Travis et al., 2016). However, MEGAN3$_{Full}$ represents a



more up-to-date and physical characterization of biogenic emissions. Future work reconciling the differences between these bottom up isoprene emissions estimates and top down constraints from measurements of composition (e.g. Kaiser et al., 740     2018) is needed.

### 7. Conclusions

We describe a novel method for simulating canopy physics relevant to atmospheric chemistry at very low computational cost. Our surrogate canopy model is based on the detailed canopy model in the MEGAN3 codebase, and simplified using a 745     statistical learning technique for the determination of variable importance. This updated scheme allows for improved physical process representation of biosphere-atmosphere interactions, including a full implementation of the MEGAN3 emissions scheme activity factors and a more direct implementation of the light and LAI dependence of dry deposition.

When implemented into a chemical transport model, this canopy scheme impacts the spatial distribution of isoprene emissions, but maintains the global total to within 5%. Consistent with prior work (Kaiser et al., 2018), isoprene emissions are reduced over the Southeast United States, with local absolute changes that can exceed 30%. This difference in surface-atmosphere exchange ultimately has a 755     modest impact on surface ozone, with absolute annual average changes generally less than 1ppbv, though it does drive ozone concentrations closer to observed values. The surrogate model additionally allows for integrating new leaf level





emission factor maps into GEOS-Chem, which we show leads to substantial changes in biogenic emissions.

In a rapidly changing earth system, it is critical to represent chemical, biological, and physical processes with as high fidelity as possible. Surrogate models that allow for rapid implementation of computationally expensive processes can play a key role in representing these processes. The work presented in this manuscript represents a step toward further explicit descriptions of biosphere-atmosphere interactions in

models of atmospheric chemistry. Future work should include more detailed observational constraints and characterization of in-canopy chemical reactions, turbulent exchange, and biological processes.

**Author Contributions**

CLH and SJS designed the study. SJS developed and implemented the surrogate

model and performed simulations and analysis. ABG developed MEGAN3 model code. All authors contributed to the manuscript preparation.

**Code Availablity**

The MEGAN3 and GEOS-Chem model code are available at:

https://bai.ess.uci.edu/megan/data-and-code, and

https://zenodo.org/record/2620535, respectively. The updated GEOS-Chem code containing the canopy model changes is available at:

https://zenodo.org/record/3614062.



**Acknowledgements**

This study was supported by the U.S. National Science Foundation (AGS 1564495)

and by NASA Headquarters under the NASA Earth and Space Science Fellowship

Program (NNX16AN92H).

**Competing Interests**

The authors declare no competing interests.

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
