# Peer review of "Development of a Reduced Complexity Plant Canopy Physics Surrogate Model for use in Chemical Transport Models: A Case Study with GEOS-Chem v12.3.0"

_Geoscientific Model Development, 2019_

## Referee Comment (RC1) · Anonymous Referee #1 · 4 Feb 2020

**Review of Silva et al., 2019**

This paper describes a surrogate canopy model based on MEGAN3 built using the LASSO regression approach. The canopy model is embedded into GEOS-Chem and compared with the default MEGANv2.1 emissions using a parameterized canopy environment (PCEEA), and a version of the full MEGAN3 model (surrogate canopy + leaf-level emission factors). In general, the surrogate canopy model reproduces leaf temperatures and PAR simulated by the full canopy model with skill. The resultant biogenic emissions are similar to those calculated using the PCEEA method. The surrogate model allows for a more complex treatment of a variety of canopy process within the GEOS-Chem. This will be an important tool for future studies of biosphere-atmosphere exchange. My comments are only minor and refer mainly to points in need of clarification:

**General comments:**

1. Clearly, the normalization factor is an important term in the calculation of global biogenic emissions. The method of calculating the standard conditions is difficult to follow (section 4.1). The following questions refer to that section:

  - Page 28, sentence starting at line 458- What does "within 10%" mean – Temperature within 10% of 303K, ranging from 273 K to 333 K? That seems very broad. Is the parenthetical comment (e.g. Current Temperature = 298.5K, Current PAR = 1500 µmol m$^{-2}$ s$^{-1}$) just an example of being within 10% of standard T and PAR? If so, stating either ranges or average values would be more helpful.

  - How sensitive are 24-hr average T and PAR to the 10% criteria, and in turn, does the normalization factor vary significantly if different criteria are used? Would use of different met fields, or a narrow geographic domain, require a recalculation of 24-hr average T and PAR, and result in different normalization factors?

  - Page 45 line 718- For "choice of standard conditions" to explain the large difference between MEGAN3$_{full}$ and MEGAN3$_{canopy}$, the normalization factor would have to be ~0.34 (=0.21*1.6). This is similar to WRF-Chem's factor (0.3). How does your model implementation of the canopy model differ? Is C$_{CE}$=0.34 within the uncertainty of your methods? How do WRF-Chem emissions compare to the values given here?

2. **Page 31, line 525.** It is unclear how "The LAI normalization in the original polynomial" differs from "direct normalization". Could you clarify the difference between the two formulations?

3. **Page 43, line 689.** "except for some updates for the contiguous US": could you be more specific or provide a reference for what these differences reflect?

**Technical comments:**

**Page 43 line 687**. "defauly" should read "default".

---

## Short Comment (SC1) · 28 Feb 2020

In your Introduction and Section 4.2 on dry deposition, you may want to discuss the recent work of Lin et al. (GBC, 2019), who showed that missing variability in ozone dry deposition velocities (Vd) from the Wesely scheme in GEOS-CHEM is due to the lack of stomatal deposition sensitivity to soil moisture deficits. Using observations at a suite of sites during wet and dry conditions, Lin et al. evaluated ozone Vd from GEOS-Chem and a new photosynthesis-based dry deposition scheme implemented in GFDL's dynamic vegetation model.

[Figure]

References:

Lin, Meiyun, Sergey Malyshev, Elena Shevliakova, Fabien Paulot, Larry W Horowitz S Fares, T N Mikkelsen, and L Zhang, October 2019: Sensitivity of ozone dry deposition to ecosystem-atmosphere interactions: A critical appraisal of observations and simulations. Global Biogeochemical Cycles, 33(10), DOI:10.1029/2018GB006157

https://agupubs.onlinelibrary.wiley.com/doi/full/10.1029/2018GB006157

---

## Referee Comment (RC2) · Anonymous Referee #2 · 16 Mar 2020

This manuscript describes the development of a surrogate canopy model for implementation in Chemistry Transport Models (CTMs). The surrogate canopy model will allow canopy physics to be represented in CTMs without increasing computational costs with the futher aim of better representing atmosphere-biosphere exchange processes, which are necessarily highly parameterized in large scale CTMs.

The authors use a machine learning regression method to establish which variables are most important for determining leaf temperature and leaf-level PAR in the canopy physics model (MEGAN3). These were identified as air temperature, and PAR incident

on the canopy and local vegetation LAI, respectively. These variables were then used in simplified models of leaf temperature and leaf-level PAR.

The surrogate canopy model was tested against MEGAN3 for leaf temperature and PAR. The surrogate canopy model was then tested against MEGAN3 in the GEOS-Chem CTM for atmospheric composition. The authors found small changes at the global scale, but large changes in the regional distribution of isoprene emissions. Finally, the authors implemented the surrogate canopy model in to GEOS-Chem in conjunction with new MEGAN3 emissions.

**General comments:**

The development of the surrogate canopy model is useful advancement in modelling biosphere-atmosphere exchange in large scale CTMs. The authors demonstrate that the surrogate canopy model performs reasonably in comparison to the canopy physics model, MEGAN3, and can improve the performance of a CTM in simulating atmospheric composition. The manuscript is generally well written and clear, and I recommend it for publication. However, I have some comments that should be addressed.

1. I felt that there could be more analysis of why the surrogate canopy model performs differently from the full canopy model, or from MEGAN(PCEEA) when implemented in GEOS-Chem. For example: - Section 3.1.1, L245-255 -> Why is the surrogate canopy model biased cool over highly vegetated/tropical regions and biased warm over northern boreal forests and arid regions?

- Section 5, L570-585 -> What exactly drives the differences in isoprene emissions between MEGAN(PCEEA) and MEGAN(Canopy) in Figures 9b and 9c? Figure 10 suggests that the differences in isoprene emissions are due to differences in the way LAI and leaf level PAR are represented in MEGAN(PCEEA) and MEGAN(Canopy), but can the authors offer any more insight in to exactly how these differences in canopy physics might drive the differences in isoprene emissions?

- Section 5, L601-621 -> Again, can the authors suggest why monoterpene emissions are less in MEGAN(Canopy) compared with MEGAN(PCEEA)?

- Section 5, L629-640 -> What drives the differences in dry deposition between MEGAN(Canopy) and MEGAN(PCEEA)

2. While I am aware that there are limited observational data sets of surface isoprene concentration, there are a number available. For example, the OP3 campaign in SE Asia (Jones et al., ACP, 2011). I think the paragraph in section 5 (L593-600) could be expanded to include more observational data sets.

3. Please re-write the captions for Figures 3-6. For example, for Figure 3: "Surrogate model performance for the annual canopy average spatial temperature in 2012. Panels...."

**Specific comments**

Abstract: L26-28 -> The authors imply that surface ozone simulated in GEOS-Chem is closer to observationally constrained values when the surrogate canopy model is used - This should be quantified with e.g. a global total.

L26-28 -> The authors state that there is no noticeable impact on computational demand - would it be useful to provide an indicative metric to illustrate this? E.g. wall clock time for a years run?

Section 3: L200-205 -> The authors calculate 20 parameters with which to model all plant canopies across the globe, ignoring the role of vegetation classes, which if considered, would increase the number of parameters to 120. Although the surrogate canopy model performs reasonably well, did the authors test if using the 120 parameters improved the performance, or conversely , degraded the computational efficiency?

L223-225 -> For clarity, could the authors adjust the following sentence to something like this: "From this relatively simple three-function parameterization (Leaf Temperature, Leaf PAR, and Sunlit Leaf fraction), we are able to implement more physically realistic parameterizations for biosphere-atmosphere interactions *in Geos-Chem/CTMs*."

Section 3.1.1 There are a couple of passages in this section that were unclear to me. Could the authors please re-word these sections so that their meaning is clearer.

L262 -> "The broad shape of the vertical distribution is consistent everywhere." By 'broad shape' do the authors refer to the canopy profile? And by 'everywhere' do the authors mean spatially, i.e. for different parts of the globe, or for different vegetation types?

L264-269 -> "The higher order variability (e.g. small differences within layers at the top and bottom of the canopy) stems from the more detailed representation of canopy energy balance in the full MEGAN3 model, which includes the influence of terms like PAR, relative humidity, LAI, and wind speed. However, the generally consistent behavior of this higher order variability allows for it to be reproduced in the simplified surrogate model."

Firstly, I do not quite understand where the 'higher order variability' occurs. In Figure 4 temperature is plotted at each canopy level so it is not possible to see any variability within canopy layers. Secondly, what is the higher order variability consistent with?

Section 3.1.2 L294-299 -> Please consider rewording the following sentences as suggested. "The low *leaf level PAR* values in the rain forest are coincident with the highest LAI values globally, leading to very strong shading effects below the first canopy layer. The northern boreal forests *have low leaf level PAR* in part due to relatively high LAI, but also due to reduced incoming PAR in the winter months when the solar angle is low."

L305-308 -> "The worst model R2 performance is over regions with the highest LAI, where the elevated importance of shading and resulting complexity in the PAR calculation is more challenging for the simplified representation of the surrogate model"

Is this really correct? Figure 5b shows the poorest surrogate model performance in

central Asia, to the west of the Andes, eastern Australia and an area of the central Sahel - all areas with low LAI according to Figure 2. I do agree that there is relatively poor surrogate model performance in the western Amazon, central sub-Saharen Africa, and perhaps Borneo (although the surrogate model performance seems ok over the rest of the maritime continent), but the statement linking poor surrogate model performance to high LAI seems too broad brush - some further clarification is perhaps needed.

L351-354 -> Does the surrogate model struggle to simulate leaf level PAR in the lower canopy for high LAI regions due to the same reasons given in L305-309, i.e. that the shading and increased complexity of the PAR calculation is harder to do in the simplified model?

**Technical comments:**

Section 4.1 L453  L459 -> Should the Current Temperature of 298.5 K in L459 be the same as the current air temperature of 303 K in L453?

Section 6 L700-701 -> Should the units of umol m-2 h-1 be consistent with the units of ug m-2 hr-1 used in Figure 14?

––––––––––––––––––––––––––––––

---

## Author Comment (AC1) · 15 Apr 2020

**Response To Reviewer Comments**

We thank the reviewers for their comments and address them in detail below.

Format: Original reviewer comments are shown in **blue text**. Our response to each comment is in **black text**.

**Reviewer 1**

Clearly, the normalization factor is an important term in the calculation of global biogenic emissions. The method of calculating the standard conditions is difficult to follow (section 4.1).

We agree with the reviewer that this section lacked important clarity. We have updated the text accordingly. Specific sub-comments are addressed below.

Page 28, sentence starting at line 458- What does "within 10%" mean – Temperature within 10% of 303K, ranging from 273 K to 333 K? That seems very broad. Is the parenthetical comment (e.g. Current Temperature = 298.5K, Current PAR = 1500 μmol m-2 s-1) just an example of being within 10% of standard T and PAR? If so, stating either ranges or average values would be more helpful.

We calculate the means jointly on all the current standard conditions. We have updated the sentences for clarity on Lines 494-498.

How sensitive are 24-hr average T and PAR to the 10% criteria, and in turn, does the normalization factor vary significantly if different criteria are used? Would use of different met fields, or a narrow geographic domain, require a recalculation of 24-hr average T and PAR, and result in different normalization factors?

The selection of candidate 24-hour average conditions is sensitive to the assumptions made in the 24-hour averaging conditional calculation. This is particularly the case when considering the geographical domain of the meteorological data, which was global in this analysis. It is certainly true that a different meteorological regime (e.g. the tropics vs boreal North America as compared to global) would produce different 24-hour average conditions. We have updated the text on lines 507-516.

Page 45 line 718- For "choice of standard conditions" to explain the large difference between MEGAN3full and MEGAN3canopy, the normalization factor would have to be ~0.34 (=0.21*1.6). This is similar to WRF-Chem's factor (0.3). How does your model implementation of the canopy model differ? Is CCE=0.34 within the uncertainty of your methods? How do WRF-Chem emissions compare to the values given here?

Our model implementation of the canopy model is different from that in WRF-Chem, as we use the updated canopy scheme from MEGAN3. The uncertainty of the biogenic emissions using the MEGAN model is at least a factor of 2, and so any differences between the model versions are within that range. We've updated the text accordingly on lines 623-625.

The WRF-Chem MEGAN emissions should be consistent with those from Guenther et al. (2012), which lead to global isoprene emissions of ~570Tg per year, consistent with MEGAN3$_{full}$.

Page 31, line 525. It is unclear how "The LAI normalization in the original polynomial" differs from "direct normalization". Could you clarify the difference between the two formulations?

We have updated the text to clarify this statement on line 579. The original polynomial decomposition in GEOS-Chem corrected for multiple normalization factors simultaneously; when the LAI normalization portion was specifically parsed out, it was found to be too high.

Page 43, line 689. "except for some updates for the contiguous US": could you be more specific or provide a reference for what these differences reflect?

The landcover and emissions data are the same as that used for MEGAN2.1 except that the landcover updates described by Yu et al. 2017 were used for the contiguous US. The updated landcover is based on high resolution (30-m) PFT and detailed vegetation types and is expected to more accurately represent the landcover distributions in this region. The text has been updated to reflect this on lines 756-759.

Page 43 line 687. "defauly" should read "default".

Thank you. Corrected.

**Reviewer 2**

I felt that there could be more analysis of why the surrogate canopy model performs differently from the full canopy model, or from MEGAN(PCEEA) when implemented in GEOS-Chem.

Thank you for the comments on where more details could be added. We updated the text accordingly, with specifics noted in response to individual comments.

Section 3.1.1, L245-255 -> Why is the surrogate canopy model biased cool over highly vegetated/tropical regions and biased warm over northern boreal forests and arid regions?

This is consistent with the removal of the vegetation class-specific traits in our simplified surrogate model. We've updated the text to reflect this on lines 262-267.

Section 5, L570-585 -> What exactly drives the differences in isoprene emissions between MEGAN(PCEEA) and MEGAN(Canopy) in Figures 9b and 9c? Figure 10 suggests that the differences in isoprene emissions are due to differences in the way LAI and leaf level PAR are represented in MEGAN(PCEEA) and MEGAN(Canopy), but can the authors offer any more insight in to exactly how these differences in canopy physics might drive the differences in isoprene emissions?

Section 5, L601-621 -> Again, can the authors suggest why monoterpene emissions are less in MEGAN(Canopy) compared with MEGAN(PCEEA)?

The MEGAN$_{PCEEA}$ and MEGAN$_{Canopy}$ parameterizations are fundamentally different models for the same process. MEGAN$_{PCEEA}$ has no vertical canopy structure and thus cannot calculate the joint impact of shading and temperature changes within a plant canopy. While the functional form for the $\gamma_T$ in MEGAN$_{PCEEA}$ is similar to MEGAN$_{Canopy}$, the $\gamma_P$ and LAI variability is completely different, and CANOPY resolves these activity factors throughout the canopy instead of in bulk as in MEGAN$_{PCEEA}$. We've updated the text on lines 549-559.

Section 5, L629-640 -> What drives the differences in dry deposition between MEGAN(Canopy) and MEGAN(PCEEA)

These differences are driven by the changes to leaf PAR and LAI normalization within the stomatal conductance algorithm in the model. The text has been updated on lines 704-705.

While I am aware that there are limited observational data sets of surface isoprene concentration, there are a number available. For example, the OP3 campaign in SE Asia (Jones et al., ACP, 2011). I think the paragraph in section 5 (L593-600) could be expanded to include more observational data sets.

We note that the bottom up emissions from the MEGAN3 scheme are directly constrained from observational datasets, so there is an implicit observational constraint on the model prior to integration. We agree with the reviewer that more detailed assessments of simulated isoprene abundances are needed. However, the nonlinear relationship between direct emissions and concentrations of isoprene and the large number of potential confounding factors (e.g. anthropogenic emissions, atmospheric oxidation capacity, model resolution, etc.) make an assessment of this sort nontrivial. The effort and scope of doing this carefully is beyond the goals of this canopy model parameterization study, but we agree it would be valuable future work. We explicitly mention this now in the conclusion section on lines 848-849.

We focus here on ozone in large part because it is more regional in nature and it has been specifically evaluated in CTMs in much greater detail (e.g. Travis et al. 2016).

Please re-write the captions for Figures 3-6. For example, for Figure 3: "Surrogate model performance for the annual canopy average spatial temperature in 2012. Panels...."

Done.

Abstract: L26-28 -> The authors imply that surface ozone simulated in GEOS-Chem is closer to observationally constrained values when the surrogate canopy model is used - This should be quantified with e.g. a global total.

We have added text to the abstract specifying the modest ozone bias reduction on lines 25-28.

L26-28 -> The authors state that there is no noticeable impact on computational demand - would it be useful to provide an indicative metric to illustrate this? E.g. wall clock time for a years run?

Thank you for the comment. We have reframed this sentence to emphasize the computational simplicity of the method, as opposed to any nebulous idea of "noticeable impact" (line 29).

Section 3: L200-205 -> The authors calculate 20 parameters with which to model all plant canopies across the globe, ignoring the role of vegetation classes, which if considered, would increase the number of parameters to 120. Although the surrogate canopy model performs reasonably well, did the authors test if using the 120 parameters improved the performance, or conversely , degraded the computational efficiency?

Using 120 parameters does slightly improve the model performance and increases the (still small) computational burden by a factor of 6. We ultimately chose not to use the 120 parameters because an increase in model complexity and free parameters by a factor of 6 was not deemed warranted given the good performance of the 20 parameter model. As a tradeoff between accuracy and model complexity we concluded that 20 parameters were sufficient, as described on lines 205-211.

L223-225 -> For clarity, could the authors adjust the following sentence to something like this: "From this relatively simple three-function parameterization (Leaf Temperature, Leaf PAR, and Sunlit Leaf fraction), we are able to implement more physically realistic parameterizations for biosphere-atmosphere interactions in Geos-Chem/CTMs."

Done.

L262 -> "The broad shape of the vertical distribution is consistent everywhere." By 'broad shape' do the authors refer to the canopy profile? And by 'everywhere' do the authors mean spatially, i.e. for different parts of the globe, or for different vegetation types?

Yes. The text has been updated on line 277.

L264-269 -> "The higher order variability (e.g. small differences within layers at the top and bottom of the canopy) stems from the more detailed representation of canopy energy balance in the full MEGAN3 model, which includes the influence of terms like PAR, relative humidity, LAI, and wind speed. However, the generally consistent behavior of this higher order variability allows for it to be reproduced in the simplified surrogate model."
Firstly, I do not quite understand where the 'higher order variability' occurs. In Figure 4 temperature is plotted at each canopy level so it is not possible to see any variability within canopy layers. Secondly, what is the higher order variability consistent with?

The higher order variability statement was unclear. We've updated the text to reflect that it was the small differences between adjacent layers on lines 280-284.

Section 3.1.2 L294-299 -> Please consider rewording the following sentences as suggested. "The low leaf level PAR values in the rain forest are coincident with the highest LAI values globally, leading to very strong shading effects below the first canopy layer. The northern boreal forests have low leaf level PAR in part due to relatively high LAI, but also due to reduced incoming PAR in the winter months when the solar angle is low."

Done.

L305-308 -> "The worst model R2 performance is over regions with the highest LAI, where the elevated importance of shading and resulting complexity in the PAR calculation is more challenging for the simplified representation of the surrogate model"
Is this really correct? Figure 5b shows the poorest surrogate model performance in central Asia, to the west of the Andes, eastern Australia and an area of the central Sahel all areas with low LAI according to Figure 2. I do agree that there is relatively poor surrogate model performance in the western Amazon, central sub-Saharen Africa, and perhaps Borneo (although the surrogate model performance seems ok over the rest of the maritime continent), but the statement linking poor surrogate model performance to high LAI seems too broad brush - some further clarification is perhaps needed.

Thank you for pointing this out. We originally did not discuss the regions with very low vegetated fractions in terms of the $R^2$ performance, as their ultimate impact on biosphere-atmosphere exchange of trace gases is low. This has been updated in the manuscript on lines 329-335.

L351-354 -> Does the surrogate model struggle to simulate leaf level PAR in the lower canopy for high LAI regions due to the same reasons given in L305-309, i.e. that the shading and increased complexity of the PAR calculation is harder to do in the simplified model?

Yes. We've included a sentence in the updated manuscript on line 383-386.

Section 4.1 L453 L459 -> Should the Current Temperature of 298.5 K in L459 be the same as the current air temperature of 303 K in L453?
Yes. Thank you, it's been corrected.

Section 6 L700-701 -> Should the units of umol m-2 h-1 be consistent with the units of ug m-2 hr-1 used in Figure 14?
Thank you for pointing this out. We've updated the text.